



# Aerosol-cloud-radiation interaction during Saharan dust episodes: The dusty cirrus puzzle

Axel Seifert[1], Vanessa Bachmann[1], Florian Filipitsch[3], Jochen Förstner[1], Christian Grams[4], Gholam Ali Hoshyaripour[4], Julian Quinting[4], Anika Rohde[4], Heike Vogel[4], Annette Wagner[2], and Bernhard Vogel[4]

[1]Deutscher Wetterdienst, Offenbach, Germany
[2]Deutscher Wetterdienst, Hohenpeißenberg, Germany
[3]Deutscher Wetterdienst, Lindenberg, Germany
[4]Institute of Meteorology and Climate Research (IMK-TRO), Karlsruhe Institute of Technology (KIT), Karlsruhe, Germany

**Correspondence:** Dr. Axel Seifert (axel.seifert@dwd.de)

**Abstract.** Dusty cirrus clouds are extended optically thick cirrocumulus decks that occur during strong mineral dust events. So far they have been mostly documented over Europe associated with dust-infused baroclinic storms. Since today's numerical weather prediction models neither predict mineral dust distributions nor consider the interaction of dust with cloud microphysics, they cannot simulate this phenomenon. We postulate that the dusty cirrus forms through a mixing instability of moist clean air with drier dusty air. A corresponding sub-grid parameterization is suggested and tested in the ICON-ART model. Only with help of this parameterization ICON-ART is able to simulate the formation of the dusty cirrus, which leads to substantial improvements in cloud cover and radiative fluxes compared to simulations without this parameterization. A statistical evaluation over six Saharan dust events with and without observed dusty cirrus shows robust improvements in cloud and radiation scores. The ability to simulate dusty cirrus formation removes the linear dependency on mineral dust aerosol optical depth from the bias of the radiative fluxes. This suggests that the formation of dusty cirrus clouds is the dominant aerosol-cloud-radiation effect of mineral dust over Europe.

## 1 Introduction

The term 'dusty cirrus' is used by meteorologists, especially in Europe for extended cirrus cloud decks that typically occur during strong Saharan dust episodes (Kollath, 2010; Roesli et al., 2020; Fierli et al., 2022). A characteristic property of this type of dusty cirrus is the cellular structure that hints at convective overturning within the cirrus cloud layer.

Dusty cirrus decks are associated with dust-infused baroclinic storms (DIBS, Fromm et al., 2016), which are large midlatitude cyclones that transport huge amounts of mineral dust from Africa to Europe. Due to the strong ascending motions in these baroclinic storms, the mineral dust can reach the upper troposphere and affect, or even cause, the formation of cirrus clouds (Ansmann et al., 2019). However, not all DIBS produce extended dusty cirrus cloud decks. It has been hypothesized that these extended dusty cirrus decks form through longwave cooling at an elevated dust layer. The longwave cooling leads to destabilization in the upper troposphere and a subsequent formation of a shallow convective cirrus cloud deck (Nagy, 2009;





Kollath, 2010). This hypothesis is largely based on the cellular structure of the cirrus cloud, which is visible in high-resolution satellite images (Figure 1). The basic mechanism is also supported by idealized numerical simulations (Fusina and Spichtinger, 2010; Spichtinger, 2014). Previous studies of cirrus clouds associated with dust events in Europe (Rieger et al., 2017; Weger

et al., 2018) or Asia (Wang et al., 2015; Caffrey et al., 2018; Pan et al., 2019) have not focused on the special characteristics of the dusty cirrus. Dusty cirrus with extended cloud decks are rare. In Europe, roughly one event per year is observed.

      Dusty cirrus clouds pose a challenge for numerical weather prediction (NWP) and climate models. Today's NWP models are in general unable to predict these dust-induced clouds. This is not surprising as operational NWP systems do currently not explicitly predict mineral dust, but employ an aerosol climatology to take into account the average effect of mineral dust

and other aerosols on radiation and cloud formation. Hence, operational NWP systems know next to nothing about the actual distribution of mineral dust in the atmosphere, especially during episodes. Global aerosol and chemistry forecasting models, like CAMS (Morcrette et al., 2009; Rémy et al., 2019) or GEOS-5 (Nowottnick et al., 2011; Bunn et al., 2020) on the other hand, do predict mineral dust, but in such models, the dust is usually not explicitly coupled to cloud formation, i.e., aerosol-cloud-interaction is not taken into account. Climate models suffer from very coarse grid spacing and the formation mechanisms

of cirrus clouds and the aerosol-cloud-interaction are therefore highly parameterized (Kuebbeler et al., 2014; Wang et al., 2014; Penner et al., 2018). A recent study suggests that climate models underestimate the aerosol-cloud-interaction for cirrus clouds (Maciel et al., 2022). Besides the effect on ice nucleation and cirrus formation, mineral dust can modify clouds and rainfall by acting as cloud condensation nuclei (e.g. Hui et al., 2008) and through a modulation of synoptic-scale and mesoscale atmospheric circulations (e.g. Jin et al., 2021; Parajuli et al., 2022).

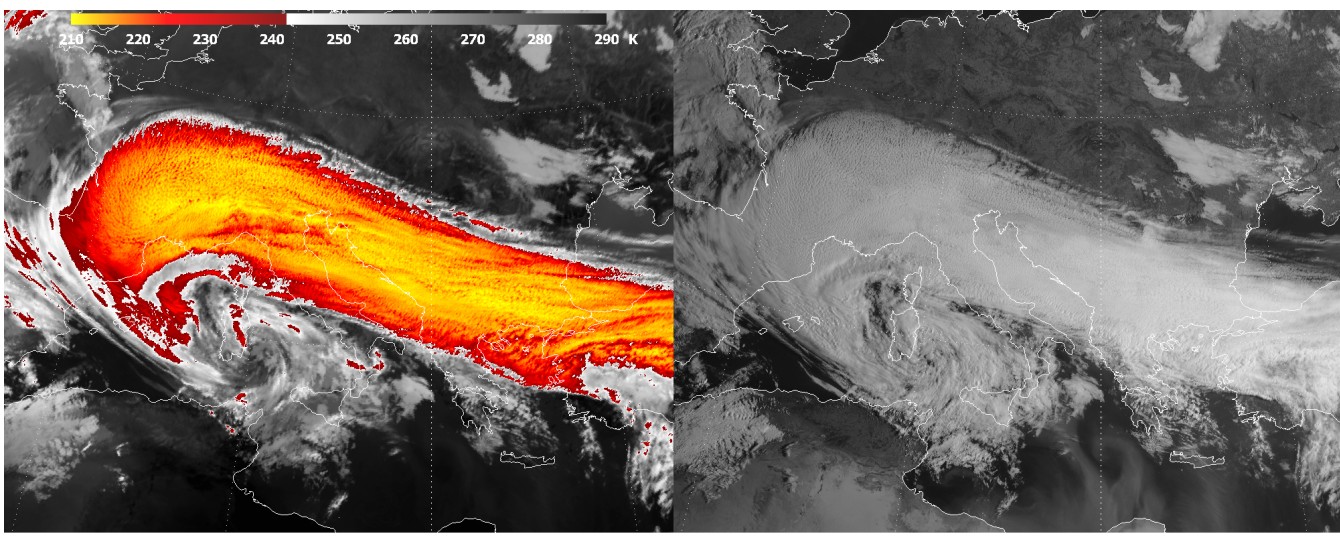

**Figure 1.** MSG SEVIRI infrared (IR) and high-resolution visible (HRV) of 21 April 2020, 6:30 UTC, with an extended dusty cirrus cloud deck over central Europe (from Roesli et al. 2020, with permission from EUMETSAT)





Not being able to predict dust-induced cirrus formation can lead to large forecast errors for the solar irradiance at the surface, even in day-ahead forecasts. The erroneous forecasts can subsequently lead to an overestimation in the prediction of photovoltaic (PV) power generation (e.g. Rieger et al., 2017). With the rising relevance of PV for the energy supply in Europe and worldwide, this poses significant challenges for system operators in the energy market and the management of the power grid itself (Antonanzas et al., 2016).

Therefore it is of high relevance to improve the NWP systems and enable them to predict these rare but important dusty cirrus cloud decks. In the following, we present a new parameterization for dusty cirrus in the model system ICOsahedral Nonhydrostatic model with Aerosol and Reactive Trace gases (ICON-ART) model, which is based on the hypothesis of a mixing instability between moist clean air with the drier Saharan dust layer. We show that the combination of explicitly predicting mineral dust with a state-of-the-art aerosol model and our new parameterization of aerosol-cloud effects leads to

skillful simulations of dusty cirrus. The ICON-ART system is then evaluated for several Saharan dust events with and without dusty cirrus occurrence.

In section 2, we introduce the ICON-ART model and the dusty cirrus parameterization, which augments the cloud scheme in the ICON model. Section 3 is dedicated to an analysis of three dusty cirrus cases, and a statistical analysis of six Saharan dust cases with and without dusty cirrus is presented. We end with a summary and conclusions.

## 2  Model description

### 2.1  ICON, ICON-ART and ICON-D2-ART

ICON is a non-hydrostatic compressible atmospheric model, which uses a triangluar icosahedral mesh (Zängl et al., 2015). ICON is developed and maintained jointly by Deutscher Wetterdienst (DWD), the Max-Planck Institute for Meteorology (MPI-M), the German Climate Computing Center (DKRZ) and the Karlsruhe Institute of Technology (KIT). The operational NWP

system at DWD consists of a global ICON model, currently at 13 km grid spacing, with a European two-way nest at 6.5 km grid spacing (called ICON-EU), and the regional ICON-D2 with approximately 2 km grid spacing over central Europe (Reinert et al., 2022). ART is a component of ICON that enables treatment of atmospheric chemistry and aerosols (Rieger et al., 2015; Schröter et al., 2018). In the following, we apply ICON-ART in a regional configuration similar to the operational ICON-D2, which we call ICON-D2-ART. The model domain has 542040 cells in each of the 65 model levels. ICON uses a vertically

stretched grid. For ICON-D2 the vertical grid spacing in the lowest levels is smaller than 100 m, but near the tropopause it is approximately 500 m (see Reinert et al., 2022, p. 121). The domain top is at 22 km height. In this study we use ART only to simulate mineral dust. The aerosol model uses a modal distribution with a two-moment formulation (Rieger et al., 2017; Gasch et al., 2017; Hoshyaripour et al., 2019). Mineral dust is represented by three log-normal modes with standard deviations of 1.7, 1.6, and 1.5 for modes dustA, dustB, and dustC, respectively. While the standard deviations are kept constant the median

diameters are variable depending on the simulated mass and number concentration. Chemical aging of dust is not taken into account in this study, although ICON-ART is in principle able to treat coated aerosol particles (Muser et al., 2020). DWD and KIT maintain a pre-operational global dust forecasting system based on ICON-ART with a grid of approximately 40 km with



vertical levels and a two-way nest with 20 km grid spacing (ICON-EU-ART). In this study we use boundary conditions for ICON-D2-ART from the ICON-EU-ART nest of the analysis cycle of this global mineral dust forecasting system.

The model physics of ICON-D2-ART is largely based on the physical parameterizations of COSMO-DE as described in Baldauf et al. (2011), but in ICON-D2-ART the two-moment mixed-phased cloud microphysics of Seifert and Beheng (2006) as described in Seifert et al. (2012) is used. The most important recent change to the two-moment microphysics is that the PDA08 ice nucleation scheme (Phillips et al., 2008) has been replaced with a parameterization of the ice nucleation active surface site (INAS) density (Ullrich et al., 2017). The INAS approach greatly simplifies the coupling of the ice microphysics

to the predicted mineral dust modes, because only the total surface area of dust is needed as an input to the ice nucleation parameterization in addition to the supersaturation. In the setup of ICON-D2-ART used in the current study, mineral dust is not depleted by ice nucleation. Therefore we use an additional tracer variable to track activated ice nuclei as described in Köhler and Seifert (2015) to avoid an overestimation of heterogenous ice nucleation. Homogeneous ice nucleation of liquid aerosols is parameterized based on Kärcher et al. (2006).

Radiative fluxes in ICON are calculated using the ecRad radiation scheme (Hogan and Bozzo, 2016, 2018; Rieger et al., 2019). ICON makes use of the Tegen et al. (1997) aerosol climatology. In the present study, ICON-D2-ART with prognostic mineral dust applies the monthly climatological values for the sulfate, organic and black carbon, and sea salt aerosol modes. Furthermore, we have reduced the sulfate, organic carbon, and black carbon aerosol optical depth compared to the original Tegen climatology to take into account the reduction of anthropogenic aerosol sources in Europe in the last decades. For

the prognostic dust, the radiative transfer parameters are calculated online depending on the optical properties and the size distributions (Hoshyaripour et al., 2019). For resolved clouds the effective radius is calculated consistent with the microphysical assumptions of the two-moment scheme. Ice optical properties of Fu (1996) are used. For effective radii larger than 100 $\mu$m, which is outside the tables currently provided by ecRad, the effective radius is rescaled with the assumption $q_i/r_\text{eff}$ =const., where $q_i$ is the ice mass fraction and $r_\text{eff}$ the ice effective radius. As an offline diagnostic the RTTOV forward operator (Saunders

et al., 2018) is used to simulate Meteosat Second Generation (MSG) observations of the spinning enhanced visible and infrared imager (SEVIRI). RTTOV is called with model consistent cloud information including the effective radii from the two-moment microphysics. For the visible channel at 0.6 $\mu$m the MFASIS (method for fast satellite image synthesis) operator (Scheck et al., 2018; Geiss et al., 2021) is applied as part of RTTOV.

   ICON does come with a diagnostic sub-grid cloud cover scheme, which is similar in spirit to the schemes of Smith (1990) and

Le Trent and Li (1991), and provides cloud fraction and sub-grid liquid and ice water content for the radiation calculation. This sub-grid cloud cover scheme is usually not explicitly coupled to the aerosol information provided by ICON-ART, i.e., aerosol-cloud-interaction happens only on the grid scale, but is neglected for sub-grid clouds. A parameterization, which changes this, is described in the following.

## 2.2   A parameterization of dusty cirrus

The sub-grid parameterization of dusty cirrus is based on the concept of a fundamental mixing instability of cold moist clean air with a drier air mass containing ice nucleating particles (INPs). During the Saharan dust events, and especially as part



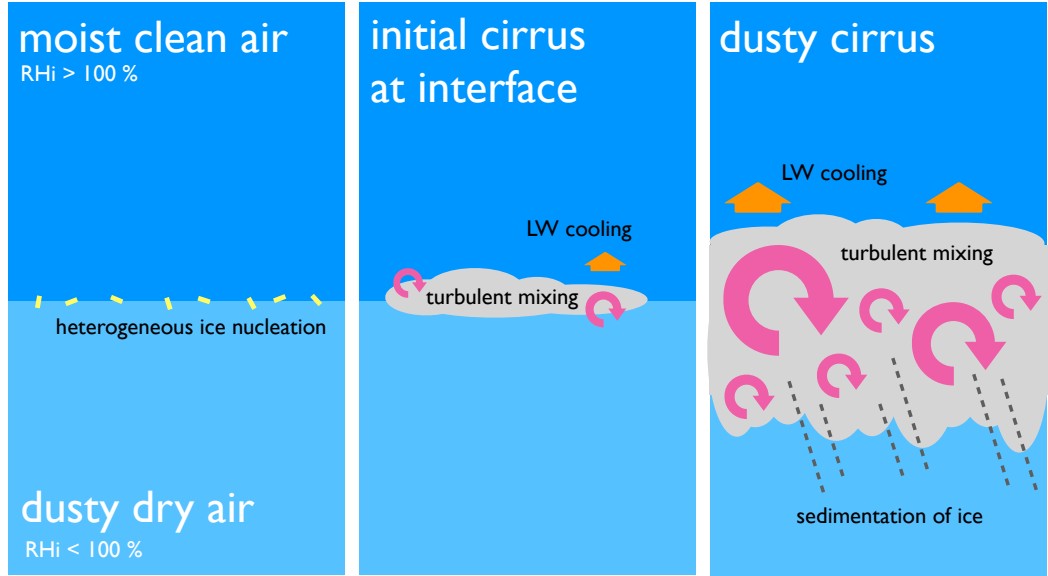

**Figure 2.** Conceptual model summarizing the physical processes that lead to the formation of the dusty cirrus at the interface between a dry Saharan dust layer and a moist atmospheric layer above. The conceptual model assumes cold environmental conditions typical for the upper troposphere and near-neutral to weakly-stable stratification.

of a dust-infused baroclinic storm (DIBS), the dynamical lifting can lead to a stratification with moist clean air in the upper troposphere located above drier dusty air below. The clean moist air has very few INPs, whereas the dusty air has INPs but lacks the moisture to form a cloud. At the interface between the two layers, heterogeneous ice nucleation can occur if the air mass has cooled to sufficiently low temperature. This can be achieved e.g. through lifting of the air layers by ascending motion underneath (Wernli et al., 2016). Small amounts of mineral dust, which are mixed into the moist air above the rather dry dust-carrying layer, can initiate the formation of an ice cloud at the interface between the two layers (see Figure 2). Longwave cooling at this thin cirrus layer can further destabilize the layer and leads to turbulent mixing. This will entrain more dust from below into the moist layer and further thicken the cirrus cloud layer. At some point, the convective overturning becomes strong enough that even homogeneous nucleation may become relevant (e.g. Spichtinger, 2014). We favor the hypothesis of a mixing instability driven by heterogeneous nucleation over the original hypothesis of longwave cooling at the dust layer postulated earlier (Nagy, 2009; Kollath, 2010), because it explains more naturally why the dusty cirrus cloud forms above rather than in the Saharan dust layer. Longwave cooling at the dust layer could play a role in the formation of the very first cirrus cloud at the moist-dusty interface, but on the other hand, the moist layer above reduces the longwave effect of the dust below. The longwave cooling at the dusty layer can contribute to the destabilization of the background profile, but lifting of the air masses, e.g. in a warm conveyor belt, leads by itself to a near-neutral or at least weakly-stable stratification in the upper troposphere (e.g. Gierens et al., 2022). Given a weakly-stable stratification, the longwave cooling at the thin ice cloud that forms at the interface between the moist clean air and the drier dusty air is sufficient to further destabilize the layer and will eventually lead





to the formation of the dusty cirrus deck. Hence, it remains unclear whether the radiative effect of the dust is important in this process. In our hypothesis, the main role of the mineral dust is to provide INPs for the cloud formation, whereas the cloud itself causes the longwave cooling that leads to the turbulent mixing of the layers and the development of the shallow convective cirrus.

To characterize the amount of dust in the Saharan dust layer, we use the mass concentration of mineral dust $c_{\mathrm{mode}}$ with mode $\in \{\mathrm{dustA, dustB, dustC}\}$. That sufficient amounts of mineral dust reach the upper troposphere is the most important predictor in our parameterization. The moisture is quantified by the ice saturation ratio $s_{\mathrm{ice}} = p_v/p_{\mathrm{sat,ice}}$, where $p_v$ is the vapor pressure and $p_{\mathrm{sat,ice}}$ is the saturation vapor pressure over ice. As a measure of atmospheric stability we use the temperature lapse rate

$$\gamma_k = \left.\frac{\partial T}{\partial z}\right|_k \approx \frac{T_k - T_{k+1}}{\Delta z} \tag{1}$$

Note that ICON uses top-down indices, i.e., level $k+1$ is below level $k$. Dusty cirrus occurs in model level $k$ if the following conditions are fulfilled:

$$T_k \quad < 240 \text{ K} \tag{2}$$

$$\hat{c}_{\mathrm{dust},k} = \max_{j=k+1}^{k+N} \left(c_{\mathrm{dustB},j} + 2\,c_{\mathrm{dustC},j}\right) > c_{\mathrm{dust}}^* \tag{3}$$

$$\hat{s}_{\mathrm{ice},k} = \max_{j=k}^{k+N} s_{\mathrm{ice},j} > s_{\mathrm{ice}}^* \tag{4}$$

$$\hat{\gamma}_k = \min_{j=k}^{k+1} \gamma_j < \gamma^* \tag{5}$$

with empirically determined thresholds $c_{\mathrm{dust}}^* = 50 \ \mu\mathrm{g \ kg}^{-3}$, $s_{\mathrm{ice}}^* = 0.7$, and $\gamma^* = -6.5 \text{ K km}^{-1}$ and with $N = 4$ corresponding to a vertical depth of approximately 1500 m. The non-locality of this parameterization corresponds to the convective overturning and the mixing instability described above.

Note that $c_{\mathrm{dust}}^*$ does not include that smallest dust mode dustA, and the largest mode dustC has double the weight of dustB. This choice helps to avoid false alarms in the prediction of the dusty cirrus. The fact that the larger dust modes, dustB and dustC, are better predictors for the occurence of a dusty cirrus than dustA is consistent with the increased ability of large mineral dust particles to act as INPs, whereas smaller particles are less relevant for the formation of ice clouds by heterogeneous nucleation (DeMott et al., 2010, 2015).

The ice saturation threshold is rather low with $s_{\mathrm{ice}}^* = 0.7$. We assume that the process starts at scales considerably smaller than the resolved scales of the model. Fluctuation due to gravity waves or shear-induced clear-air turbulence can locally initiate the cirrus cloud formation at the air mass interface. Once an initial cirrus layer has formed, this triggers the microphysical mixing instability as described above.

The dusty cirrus is further characterized by a cloud fraction of one, $C = 1$, a maximum ice water content of $\mathrm{IWC}_{\mathrm{dusty}}^* = 50$ mg m$^{-3}$ and an ice particle number density of $500 \times 10^{-3}$ m$^{-3}$. For the ice water content of the dusty cirrus in a model level $k$ we apply a linear tapering with

$$\mathrm{IWC}_{\mathrm{dusty},k} = \mathrm{IWC}_{\mathrm{dusty}}^* \max\left\{\min\left[1.0, \frac{\hat{c}_{\mathrm{dust},k} - c_{\mathrm{dust}}^*}{c_{\mathrm{dust}}^\circ - c_{\mathrm{dust}}^*}\right], 0.1\right\} \max\left\{\min\left[1.0, \frac{\hat{s}_{\mathrm{ice},k} - s_{\mathrm{ice}}^*}{s_{\mathrm{ice}}^\circ - s_{\mathrm{ice}}^*}\right], 0.1\right\} \tag{6}$$





**Table 1.** Overview of ICON-D2-ART simulation periods. The initial condition is at 00 UTC on the initial date. The table gives the maximum mineral dust AOD of ICON-D2-ART. The occurrence of dusty cirrus is estimated based on satellite data.

| Initial date | Time period | max. dust AOD | dusty cirrus |
| --- | --- | --- | --- |
| 15 March 2022 | 15-19 March | 1.00 | yes |
| 1 March 2021 | 1-5 March | 0.58 | yes |
| 4 May 2022 | 4-8 May | 0.68 | yes |
| 21 Feb 2021 | 21-25 Feb | 1.23 | no |
| 27 April 2022 | 27 Apr - 1 May | 0.54 | no |
| 18 June 2021 | 18-22 June | 0.69 | no |

with $c_{\text{dust}}^{\circ} = 70 \ \mu g \ kg^{-3}$, $s_{\text{ice}}^{\circ} = 1.0$.

According to our hypothesis and consistent with observations, convective overturning on scales smaller than 10 km is a crucial ingredient for the formation of the dusty cirrus cloud deck. Given the horizontal grid spacing of 2 km, with an effective resolution larger than 10 km, and a vertical grid spacing coarser than 500 m in the upper troposphere, it is reasonable that
ICON-D2-ART is not able to explicitly simulate the chain of processes that leads to the formation of dusty cirrus. It seems therefore appropriate to describe the dusty cirrus by a parameterization as part of the sub-grid cloud scheme as formulated above.

## 3 ICON-D2-ART simulations of Saharan dust events

In this study, six Saharan dust episodes over Europe are evaluated. Each simulation spans a time period of five days and is
initialized from the analysis cycle of the global dust forecasting system of DWD, which also provides the lateral boundary conditions. For each dust episode, an ICON-D2 simulation without prognostic dust acts as a control run mimicking the behavior of a standard operational NWP system ('no dust'). In addition, three ICON-D2-ART simulations are discussed. First, a simulation with only aerosol-radiation interaction (ARI). Second, a simulation with ARI and grid-scale aerosol-cloud-interaction based on the two-moment microphysics scheme (ACI). Third, a simulation with aerosol-radiation interaction, grid-scale aerosol-cloud-
interaction, and the sub-grid dusty cirrus parameterization (ACI-dusty). The simulation periods are summarized in Table 1 and the differences between the simulations are detailed in Table 2. In the following we discuss the three dusty cirrus events. Detailed information about all six simulation periods is provided in the Supplement to this paper.

### 3.1 Dusty cirrus case of 15-19 March 2022

On 15 March 2022, 00 UTC, the center of the storm 'Celia' is located west of the Strait of Gibraltar. In the mid- to upper-
troposphere it is associated with a deep trough that reaches equatorward as far as the western Sahara. Located south of a quasi-stationary anticyclone over Europe, the trough decouples from the westerly flow and forms a cut-off on 16 March 2022. On the eastern flank of the trough in a region of quasi-geostrophic forcing for ascent, large amounts of Saharan dust are





**Table 2.** Overview of ICON-D2-ART simulations performed for each dust episode.

| Simulation name | climatological dust | prognostic dust | aerosol-radiation interaction | aerosol-cloud interaction | dusty cirrus parameterization |
|---|---|---|---|---|---|
| no dust | ✓ | - | - | - | - |
| ARI | - | ✓ | ✓ | - | - |
| ACI | - | ✓ | ✓ | ✓ | - |
| ACI-dusty | - | ✓ | ✓ | ✓ | ✓ |

transported poleward towards Spain and lifted from the lower into the mid- to upper troposphere. Over the next two days, the dusty air mass moves further eastward over France and Germany with a dust aerosol optical thickness (AOD) exceeding 0.8 in

the ICON-D2-ART simulation (Figure 3).

The dusty air mass is associated with an extended cirrus cloud deck, which covers most of France and Switzerland on 16 March, 12 UTC, as can be seen in the Meteosat SEVIRI visible image shown in Figure 4a (left). The infrared brightness temperature from SEVIRI reaches 215 K over large parts of the cirrus cloud deck (4b; left). The ACI simulation of ICON-D2-ART shows clear-sky conditions over most of France and Switzerland, and mid-level clouds with brightness temperatures

larger than 240 K in the northern part of France (4a,b; center). This simulation fails to simulate the cirrus associated with the Saharan dust, although the simulated dust AOD matches the spatial extension of the cirrus cloud. This changes with the sub-grid dusty cirrus parameterization in the ACI-dusty simulation, which enables ICON-D2-ART to simulate the cirrus cloud deck that is associated with the Saharan dust (4a; right). The infrared brightness temperature is somewhat too low for 16 March 12 UTC, but the spatial extent agrees well with the SEVIRI observations (4b, right cf. left). On 17 March 00 UTC, when cirrus

clouds extend over the Netherlands, the North Sea, Denmark, and most of Germany we find a good agreement between SEVIRI and the ACI-dusty simulation (4c). The purely grid-scale ACI simulation is not able to simulate the cirrus clouds and shows only low and mid-levels clouds.

An overpass of the Aqua satellite operated by NASA provides CERES radiative flux data for 16 March at 12:30 UTC as shown in Figure 5. Throughout this study the CERES SSF Level 2 Edition-4A instantaneous fluxes are used (Su et al., 2015a, b;

Kratz et al., 2020). The cirrus cloud deck over France leads to reduced outgoing longwave flux and a large reflected shortwave flux at top of atmosphere (TOA). The solar irradiance at the surface is greatly reduced to values below 200 $\mathrm{W\,m^{-2}}$ under the cirrus cloud deck. Consequently, the ACI simulation without the cirrus cloud deck overestimates the shortwave surface flux and the outgoing longwave radiation (OLR) and underestimates the reflected shortwave at TOA. The errors in the solar irradiance exceed a factor of 2 over large areas in the ACI simulation. In the ACI-dusty simulation with the sub-grid dusty

cirrus scheme, the cirrus cloud deck is properly represented and all three radiative fluxes are reasonably consistent with the CERES observations.

To quantify the errors in the radiative fluxes, we make use of all CERES data from Aqua and Terra overpasses over the ICON-D2-ART domain during the five-day period. These are a total of 35 overpasses, but only 17 during daytime with non-zero shortwave data. The CERES data is processed in 20 min time windows with radiative flux output from ICON-D2-ART every





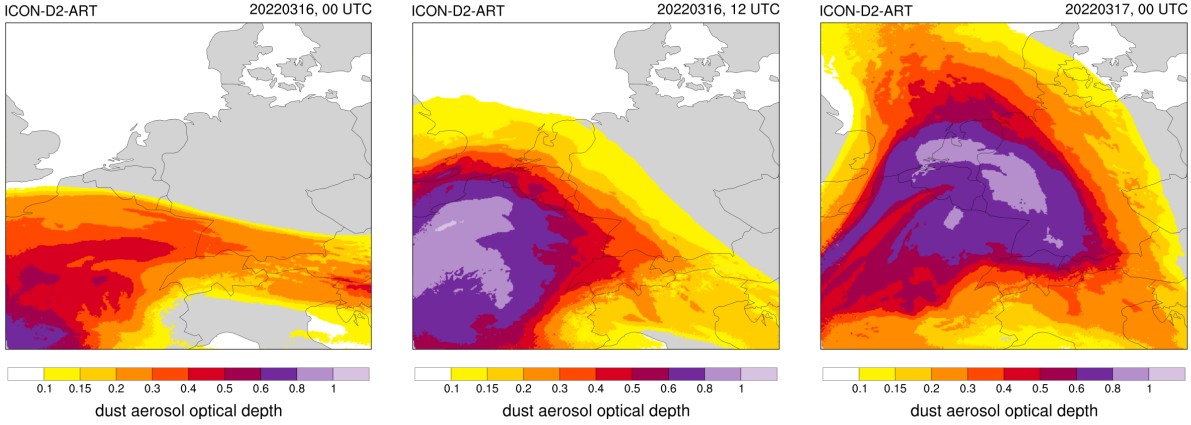

**Figure 3.** ICON-D2-ART mineral dust aerosol optical thickness (AOD) on 16 March 2022, 00 and 12 UTC, and 17 March 00 UTC.

20 min. Based on this data the biases and mean absolute errors (MAE) of the ICON-D2-ART simulations can be calculated using each single CERES SSF footprint of 25 km diameter. For each CERES footprint, 64 ICON triangles around its center are averaged. Figure 6 shows the result stratified by clear-sky/clean, clear-sky/dusty, cloudy/clean and cloudy/dusty conditions. Clear-sky is defined as both, observation and model, having a cloud cover smaller than 5 %. As observed cloud cover, the MODIS cloud cover as provided with the CERES SSF data is used. Clean and dusty pixels are defined solely based on ICON-

D2-ART because MODIS does not provide an AOD for cloudy pixels. The dust AOD threshold for dusty pixels is set to 0.1. Due to the explicit representation of the direct aerosol-radiation effect, we would expect that all three ICON-D2-ART simulations with prognostic mineral dust can improve the radiative fluxes in clear-sky/dusty conditions, but show little changes in the clear-sky/clean situations. This is confirmed for the shortwave surface flux in Fig. 6c, but only a minor improvement is seen in the TOA fluxes. For cloudy/dusty pixels the ARI and ACI simulations show an improvement over the control run, but

this is most likely also due to the direct aerosol effect because no significant difference exists between ARI and ACI. In fact, ACI with grid-scale aerosol-cloud-interaction is slightly worse than the ARI simulation. The dusty cirrus parameterization used in the ACI-dusty simulation greatly improves all three radiative fluxes for cloudy/dusty pixels and also in the all-sky statistics. Especially the biases are dramatically reduced for ACI-dusty. The MAE is for all simulations rather high in cloudy situations because it suffers from a double-penalty problem and the fact the clouds are far from perfect on small scales, but the reduction

of MAE in the ACI-dusty simulation is nevertheless a great improvement over the other three simulations.

To gain some understanding of the vertical structure of this dusty cirrus event, Fig. 7 shows a comparison of the radiosonde measurement at Payerne, Switzerland, on 16 March 12 UTC with the corresponding profiles from ICON-D2-ART. The skew-T-log-p diagram for the ACI simulation (Fig. 7a) shows a good agreement for the temperature profile and also for dewpoint temperature, except for a layer between 250 and 375 hPa in which the dewpoint temperature of the ACI simulation is 10 K too

low. Initially, we thought that the lack of moisture in the layer in which the dusty cirrus is observed is the reason that the ACI simulation fails to predict the cirrus cloud deck. All attempts to track down this error, for example, in the initial or boundary





a) visible reflectance, 16 March 2022, 12 UTC

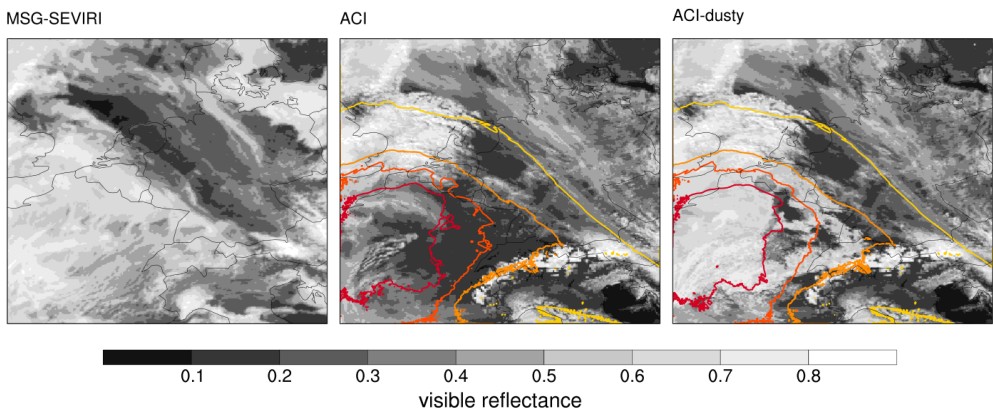

b) infrared brightness temperature, 16 March 2022, 12 UTC

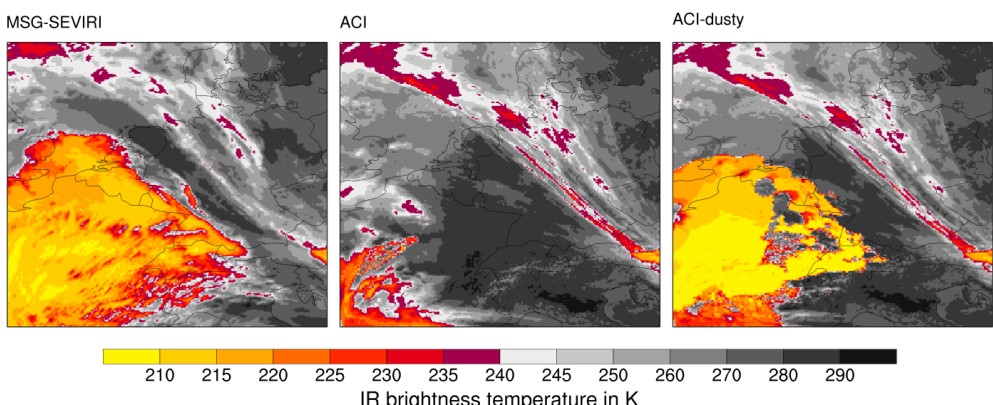

c) infrared brightness temperature, 17 March 2022, 5 UTC

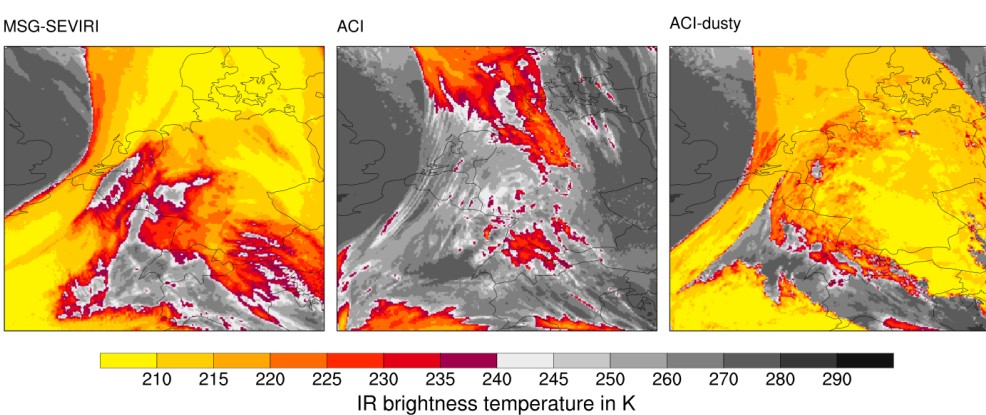

**Figure 4.** MSG SEVIRI visible reflectance (top) and infrared brightness temperature of the 10.8 $\mu$m channel (center) vs ICON-D2-ART on 16 March 2022, 12 UTC, and infrared brightness temperature on 17 March 2022, 5 UTC (bottom). Shown are MSG SEVIRI (left) vs ICON-D2-ART ACI (center) and ACI-dusty (right) simulations. The isolines in (a) are the mineral dust AOD predicted by ICON-D2-ART (interval of 0.2 starting at 0.1)



a) solar irradiance at the surface, 16 March 2022, 12:30 UTC

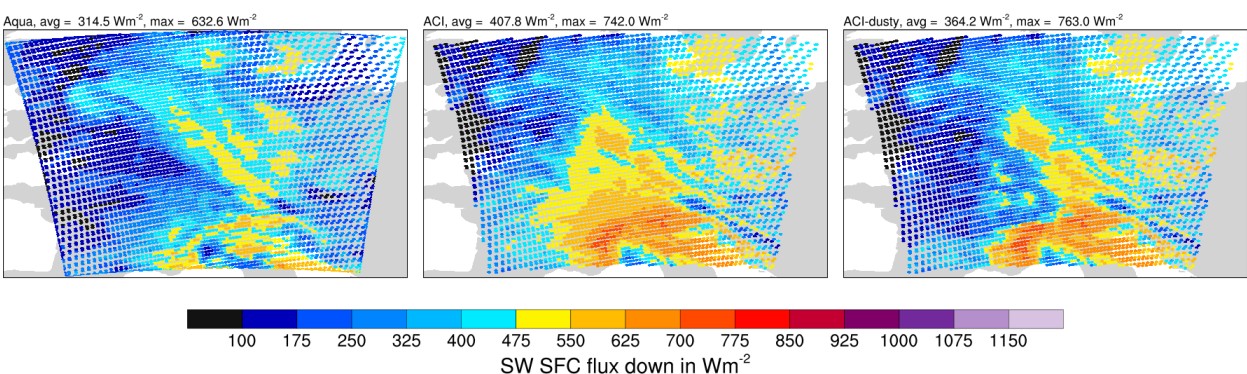

b) reflected shortwave radiation at top of atmosphere, 16 March 2022, 12:30 UTC

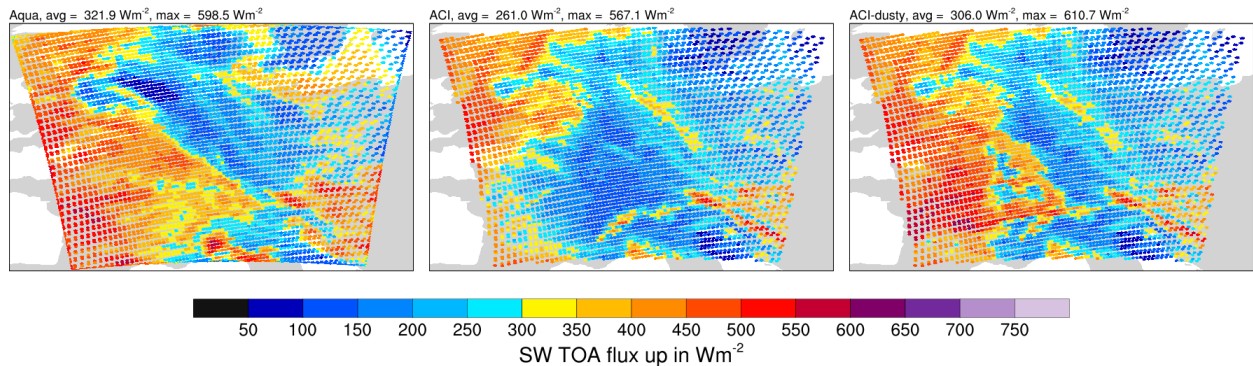

c) outgoing longwave radiation at top of atmosphere, 16 March 2022, 12:30 UTC

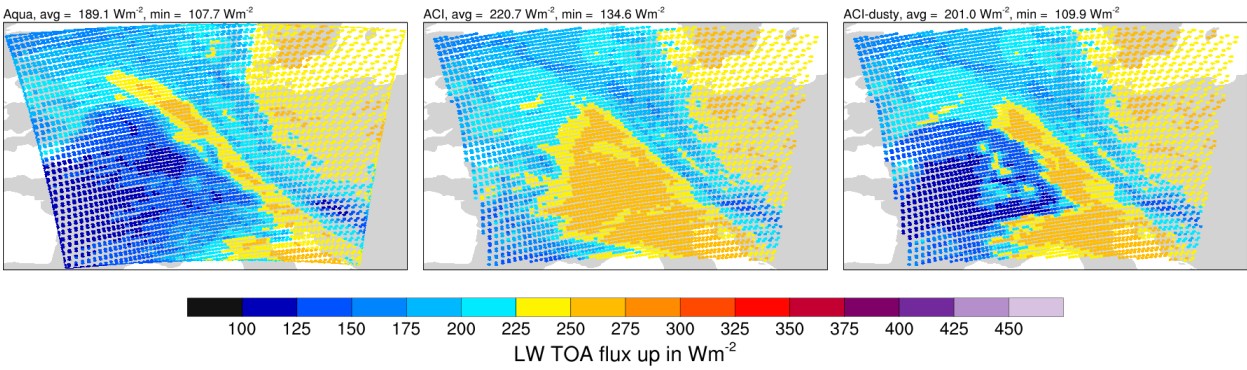

**Figure 5.** CERES SSF radiative fluxes from Aqua (left) vs ICON-D2-ART ACI (center) and ACI-dusty (right) simulations for 16 March 2022, 12:30 UTC.

a) ICON-D2-ART vs CERES outgoing longwave radiation at TOA

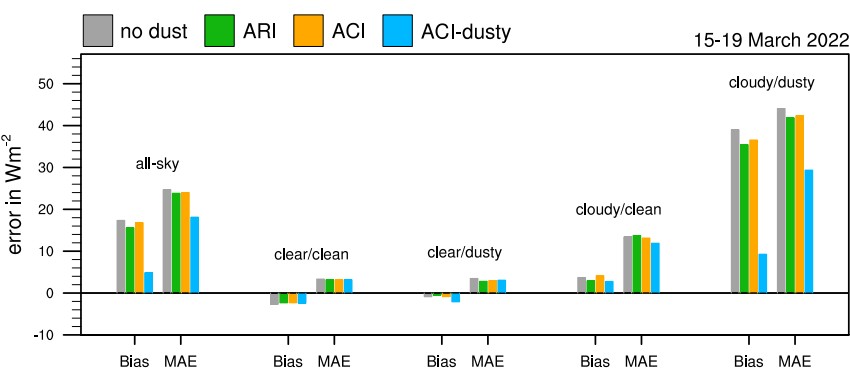

b) ICON-D2-ART vs CERES reflected shortwave radiation at TOA

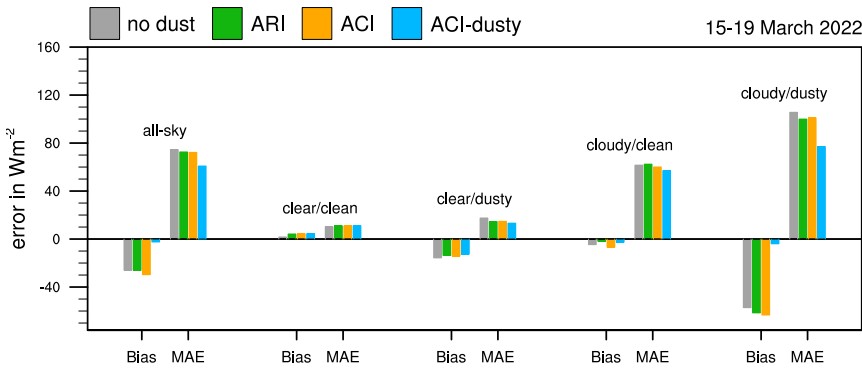

c) ICON-D2-ART vs CERES downward shortwave radiation at the surface

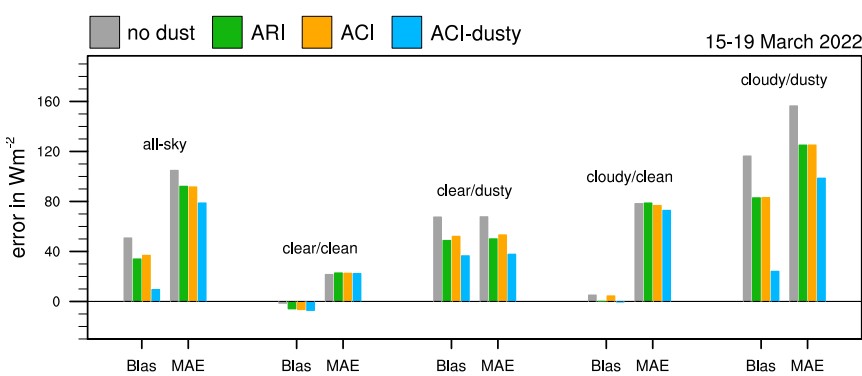

**Figure 6.** Bias and mean absolute error (MAE) of ICON-D2-ART compared to CERES SSF level 2 radiative fluxes for 15-19 March 2022. Shown are the outgoing longwave radiation at the top of atmosphere (TOA), the reflected shortwave at TOA and the solar irradiance at the surface (from top to bottom)





conditions proved to be futile, though. Interestingly, the ACI-dusty simulation corrects this error in the dewpoint temperature and shows a good agreement for the whole profile. This is even more remarkable as the dusty cirrus scheme itself does not directly modify the (grid-scale) moisture profile. As described in the previous section, it is a sub-grid cloud scheme that affects
only the radiative fluxes. Hence, the only possible explanation is that the longwave cooling at the parameterized sub-grid dusty cirrus changes the dynamics and the moisture transport in the model in such a way that it becomes much more consistent with the observations.

Another set of variables for the same sounding at Payerne is shown in Figs. 7c,d. Here we focus on the vertical profiles of the ice saturation ratio and the dust concentration. Although standard radiosondes do not measure cloudiness or ice water
content, the ice saturation ratio allows us to identify the location of the dusty cirrus in the observations. Due to the high IWC, the dusty cirrus layer is close to ice saturation, i.e. $s_{ice} = 1$. The ACI simulation underestimates the ice saturation ratio in the dusty cirrus layer and is consequently not able to simulate any cloud formation. The sub-grid dusty cirrus parameterization represents the dusty cirrus as a cloud cover of one (green dashed line in Fig. 7d). To make the connection with the SEVIRI data, the infrared brightness temperature from SEVIRI and RTTOV forward calculation is depicted by a small blue cloud symbol
(darker colors represent the observations). For optically thick clouds this can be identified with the cloud top pressure. First, this sounding confirms the concept of moist air located over drier dusty air, i.e., the dusty cirrus forms initially above not in the dust layer. Second, it shows that the dusty cirrus layer of ACI-dusty is consistent with the observed profile, and the cloud top is consistent with SEVIRI. Third, it is remarkable that the simulated ice saturation ratio increases and reaches almost one in the dusty cirrus layer, although the dusty cirrus parameterization does not directly change the water vapor. That the increase
in moisture has dynamical causes is consistent with the fact that the dust profiles change significantly as well, when the dusty cirrus parameterization is employed. Compared to the ACI simulation the ACI-dusty case has a stronger vertical transport of mineral dust and the dust mixes more efficiently with the moist layers above. This is consistent with our conceptual model but was not necessarily expected from this very simple sub-grid parameterization that does not explicitly modify the vertical transport of dust or moisture.

This behavior is also confirmed by the radiosonde profile at Lindenberg (Germany) from 17 March, 5 UTC (Figure 8). The observed sounding shows as cirrus cloud layer between 200 and 300 hPa at ice saturation, similar to the Payerne sounding. The maximum of the dust layer is here at 300 hPa and no longer separated from the cloud layer. Hence, an observation like this could be interpreted as if the cloud has formed in the dust layer. The comparison with the earlier sounding at Payerne suggests, though, that the Lindenberg sounding is characteristic of the mature state of the dusty cirrus when the mixing of the moist layer
and the dusty layer has already happened.

To substantiate the concept of moist clean air above drier dusty air prior to the dusty cirrus formation, we calculate five-day kinematic backward trajectories with the Lagrangian analysis tool (Lagranto; Sprenger and Wernli, 2015). The underlying data are 3-hourly ERA5 reanalysis (Hersbach et al., 2020) on model levels and a regular 0.5x0.5° latitude-longitude grid. The backward trajectories are started from an equidistant grid of 25 km spacing surrounding Payerne (6–10°E; 46–49.5°N) and
every 50 hPa between 450 and 200 hPa. The levels between 450–350 hPa represent the dust layer and levels between 300-200 hPa represent the moist layer above (cf. Figure 7c,d). The trajectory analysis reveals that the bulk of air parcels in the moist layer



a) Skew-T diagram ACI simulation

b) Skew-T diagrams ACI-dusty simulation

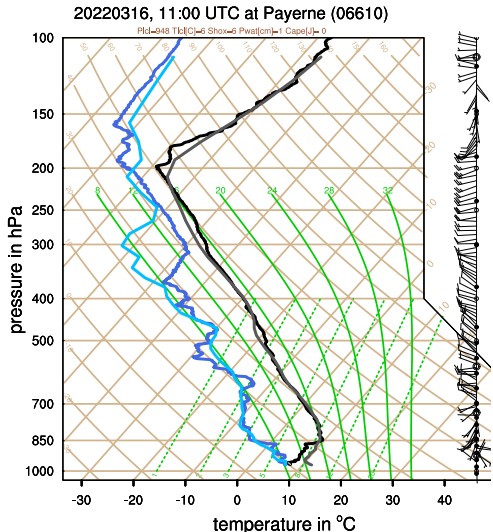

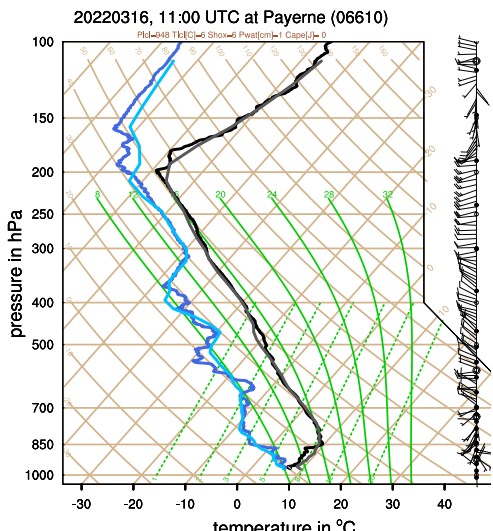

c) Vertical profiles from ACI simulation

d) Vertical profiles from ACI-dusty simulation

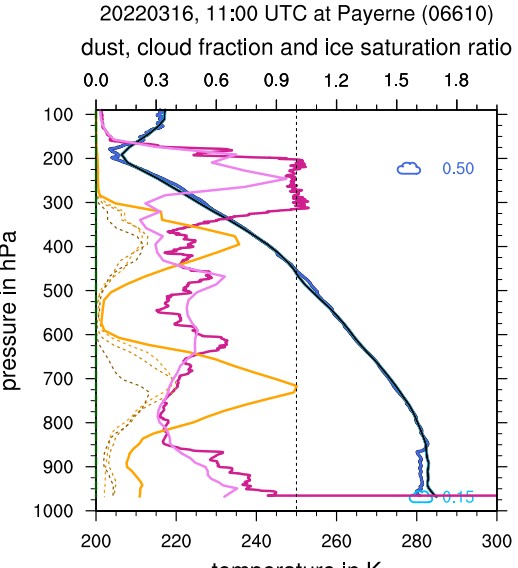

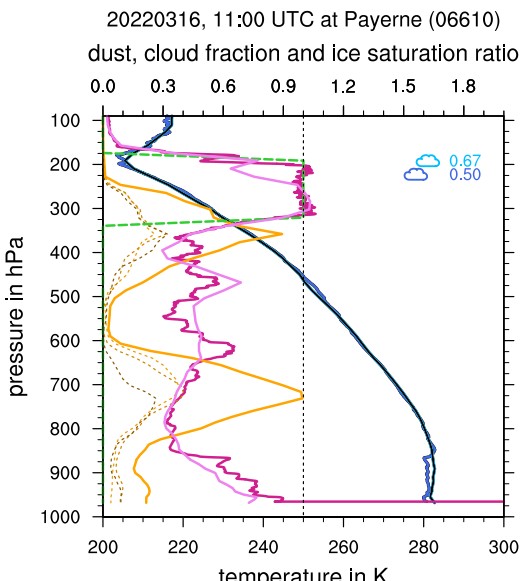

**Figure 7.** Vertical profiles from radiosonde measurements at Payerne (Switzerland) on 16 March 2021 11 UTC and the corresponding ICON-D2-ART profiles. Skew-T-log-p diagrams (a,b) showing temperature and dew point temperature from observations (black and dark blue) and ICON-D2-ART (grey and light blue). Lower plot show temperature from observations (dark blue) and ICON-D2-ART (blue), ice saturation ratio (obs: dark pink, model: light pink) and total dust concentration (orange solid) and dust modes (dashed, dustA: light orange, dustB: orange, dustC: dark orange). Dust concentrations are normalized with a constant value of $100\ \mu\mathrm{g\,m}^{-3}$. Cloud symbols indicate pressure height where IR brightness temperature (SEVIRI: dark blue, ICON-D2-ART: light blue) matches temperature of the sounding, numbers next to it indicate visible reflectance. The green dashed line is the cloud fraction predicted by ICON-D2-ART.





a) Vertical profiles from ACI simulation

b) Vertical profiles from ACI-dusty simulation

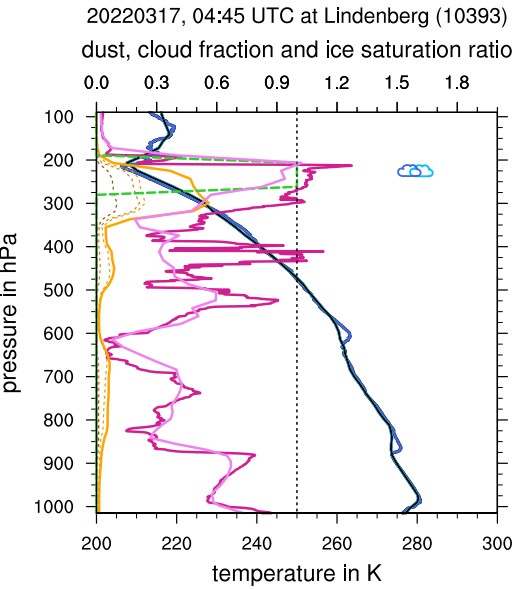

**Figure 8.** As Fig. 7c,d but for radiosonde measurements at Lindenberg (Germany) on 17 March 2021 5 UTC and the corresponding ICON-D2-ART profiles.

reaching Payerne on 16 March, 12 UTC ascended almost five days earlier over the Carribean and were transported eastward with the midlatitude jet (Figure 9a). Humidity in this layer was anomalous high, reflected by values of specific humidity in the upper 10% of the climatological distribution in the month of March based on ERA5 (not shown). The characteristic U-shape

of the trajectories over the eastern North Atlantic shows their transport around the upper-level trough associated with storm ,Celia'. As already indicated in the synoptic description, the air masses were lifted from 450 to 250 hPa ahead of the upper-level trough 36 to 18 hours (between 15 March, 00 to 18 UTC) before arriving over Payerne (Figure 10a, and triangles in 9a showing air parcel locations 24 hours prior to arriving over Payerne). We note, that this lifting of the moist layer also occurred spatially in the region of initial dusty Cirrus formation over France. Since potential temperature remained nearly constant

during this ascent, we conclude that the lifting was mostly dry adiabatic, and – as discussed in the following – initiated through the injection of the dusty layer underneath via an ascending air stream. Concerning the dry dusty layer, the majority of air parcels ascended from the lower troposphere over North Africa during the 5-day period (Figure 9b). Several air parcels start below 800 hPa and reach above 400 hPa few days later reflecting ascent in a warm conveyor belt (WCB; Wernli, 1997) in the warm sector of the forming cyclone. Only a smaller fraction, probably reminiscent of the moist layer, ascended already over

the Carribean. About 36 to 24 hours prior to reaching the region of Payerne, the air parcels ascended on average from 600 to above 400 hPa, accompanied by latent heat release, reflected by an increase of potential temperature (10b). This coincides with the lifting of the moist layer above and suggests, that WCB ascent also lifted the air column above, corroborating the notion of two distinct air layers and the formation of a cirrus cloud deck through lifting and thereby adiabatic cooling and the injection





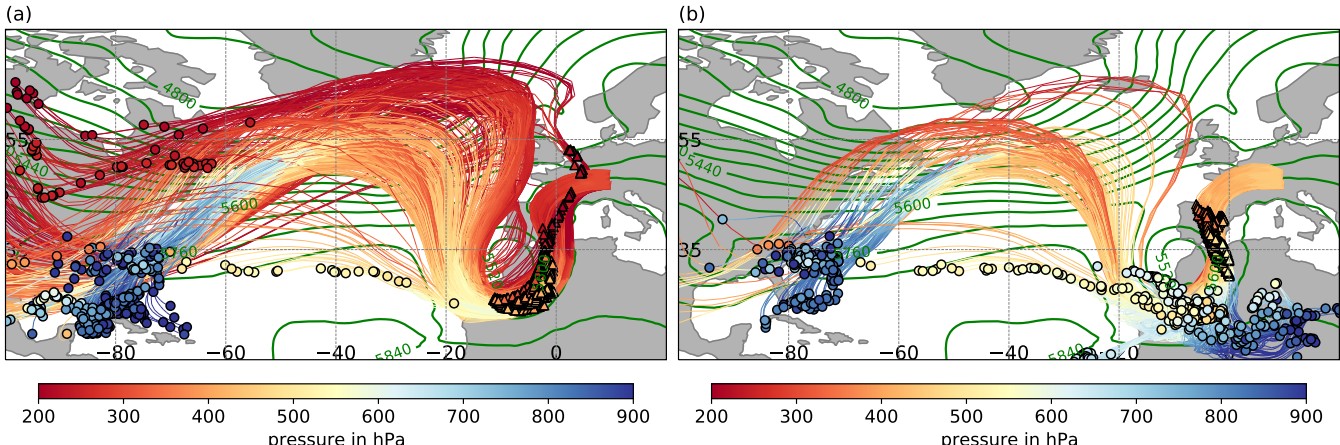

**Figure 9.** 5-day backward trajectories representing (a) the moist layer and (b) the dust layer in the Payerne region on 16 March, 12 UTC. Trajectories, air mass locations at -120 h (circles), and air mass locations at -24 h (triangles) are coloured according to their pressure height (see colorbar). Green contours show 500-hPa geopotential height in gpm at -24 h, i.e. on 15 March, 12 UTC.

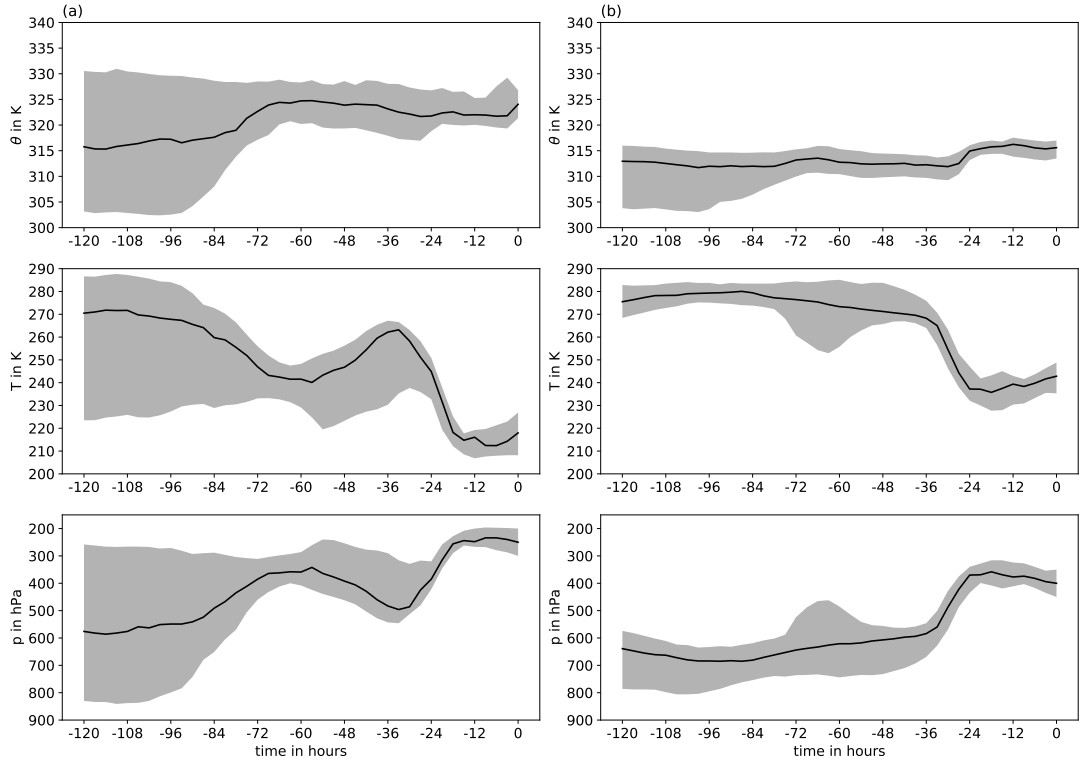

**Figure 10.** Temporal evolution of potential temperature (TH), temperature (T) and pressure (p) along backward trajectories representing (a) the moist layer and (b) the dust layer in the Payerne region on 16 March, 12 UTC. Line shows median of all trajectories and shading denotes the interquartile range.





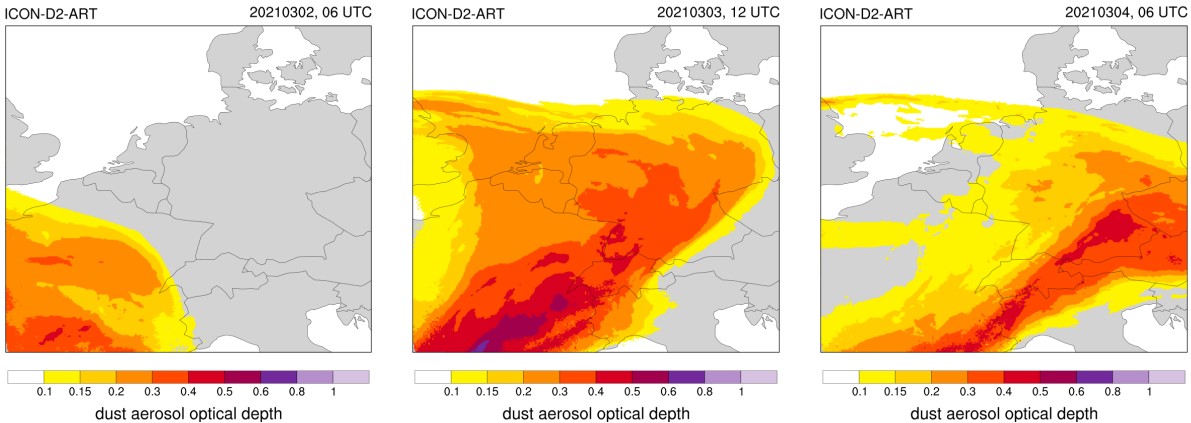

**Figure 11.** ICON-D2-ART mineral dust aerosol optical thickness (AOD) on 3 March 2021, 06 and 12 UTC, and 4 March 06 UTC.

of ice condensation nuclei into the moist layer (cf. Wernli et al., 2016). Our main conclusion is therefore that the lifiting of

an anomalously upper-tropospheric moist air layer originating from the Caribbean through a synoptic-scale ascending WCB airstream underneath emerging from the Sahara and the associated injection of dust particles as ice condensation nuclei at the interface of the moist and dusty layers have created the conditions for the formation of the dusty cirrus cloud deck. Although a climatological investigation is beyond the scope of this study, we hypothesize that the complex interaction of the various components (upper-level moist layer, synoptic forcing, dust transport in WCB) makes dusty cirrus events so rare.

## 3.2 Dusty cirrus case of 1-5 March 2021

On 2-3 March 2021, the synoptic situation is similar to the case of 16 March 2022, with a trough in 500 hPa extending from the British Isles over Spain to Marocco. Southerly flow at the eastward flank of the trough advects mineral dust directly from the Saharan Desert and Northern Africa to Europe. The event is not as massive as the 16 March 2022, but dust AODs simulated by ICON-D2-ART do exceed 0.5 over Southern France on 3 March 2021, 12 UTC (Figure 11) and high dust AOD extends along

a north-east line into Switzerland and Southern Germany.

The MSG-SEVIRI images for 3 March 2021, 12 UTC, show a cirrus cloud band associated with the Saharan dust event, which agrees well with the simulated dust plume of ICON-D2-ART (Figure 12). As in the previous case, the ACI simulation fails to predict the dusty cirrus and shows clear-sky conditions instead (including the snow-covered Alps). The ACI-dusty simulation does have the cirrus cloud band, but the boundaries of the cloud deck are too sharp and the infrared brightness tem-

perature is too homogeneous compared to the SEVIRI observations. These are probably deficiencies of the simple diagnostic parameterization that can not capture any transient behavior during the formation and decay of the dusty cirrus. Nevertheless, the sub-grid parameterization clearly improves the simulation of this dusty cirrus event by ICON-D2-ART. These improvements are also seen in the comparison to the CERES SSF data from the Aqua overpass of 3 March 2022, 11:55 UTC (Figure 13). Whereas the ACI simulation completely misses the radiative signature of the dusty cirrus cloud deck, the ACI-dusty





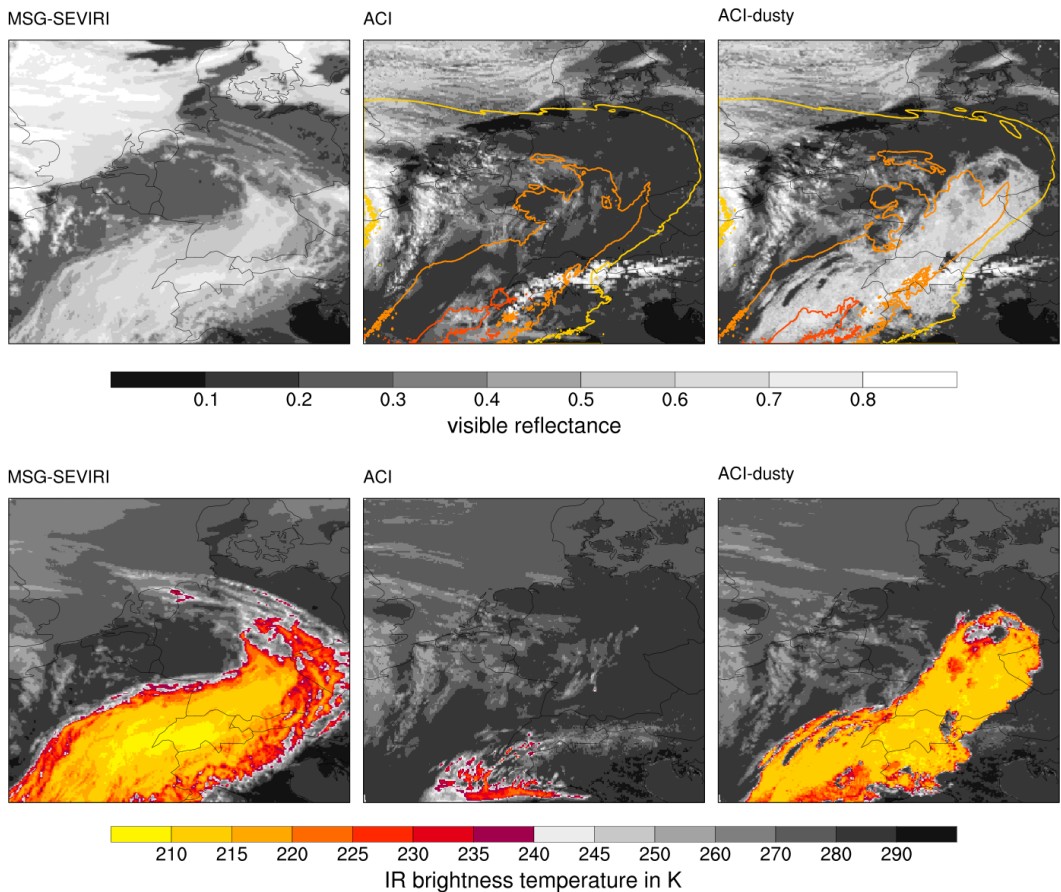

**Figure 12.** MSG SEVIRI visible reflectance (top) and infrared brightness temperature of the 10.8 $\mu$m channel (center) vs ICON-D2-ART on 3 March 2021, 12 UTC. The isolines are the mineral dust AOD predicted by ICON-D2-ART (interval of 0.2 starting at 0.1)

matches the CERES data remarkably well in all three radiative fluxes. For the broadband fluxes the tapering of the IWC of Eq. (6) seems to work better than for the RTTOV forward simulation of the MSG SEVIRI channels. Both simulations have problems to properly represent the low clouds over the North Sea and England, but this is most likely not related to Saharan dust. For the validation of the vertical distribution of cloud occurrence, we can also make use of ceilometer data for this event. Results show a significant improvement of the modelled vertical cloud fraction for the ACI-dusty simulation compared to the

other simulations (see Supplement for details).

    The good agreement of ACI-dusty with the CERES SSF fluxes is also confirmed by the scores for all Aqua and Terra overpasses from 1-5 March 2021 shown in Figure 14. For this time period all three ICON-D2-ART simulations with prognostic dust show a clear improvement in bias and MAE of the shortwave fluxes for clear-sky/dusty pixels. For cloudy/dusty pixels the ACI-dusty simulation drastically improves the representation of the three radiative fluxes. For all-sky conditions the bias of





**Figure 13.** CERES SSF from Aqua vs ICON-D2-ART for 3 March 2021, 11:55 UTC.

a) ICON-D2-ART vs CERES outgoing longwave radiation at TOA

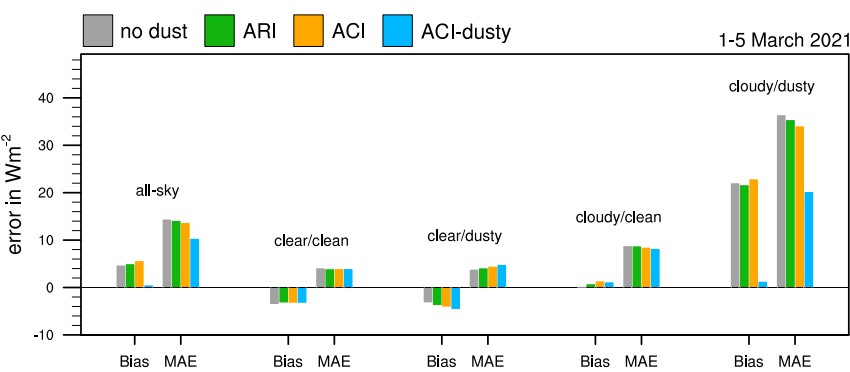

b) ICON-D2-ART vs CERES reflected shortwave radiation at TOA

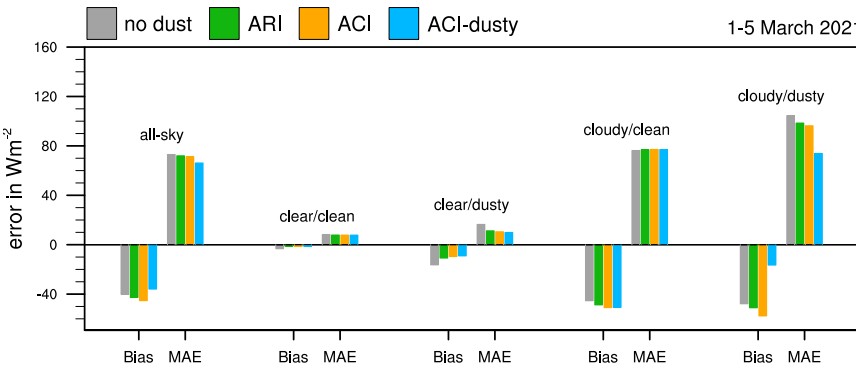

c) ICON-D2-ART vs CERES downward shortwave radiation at the surface

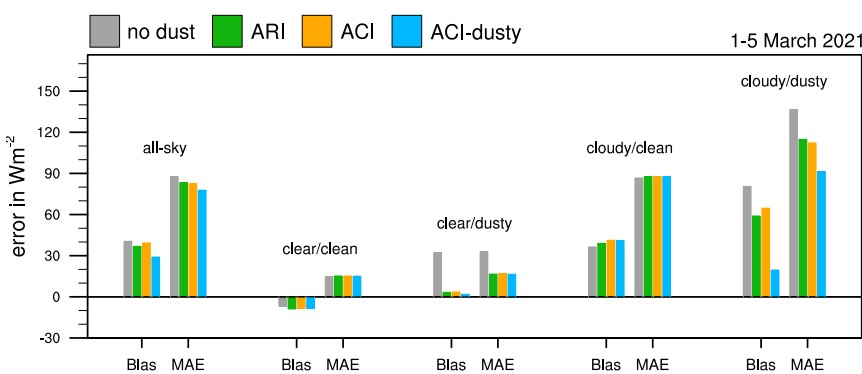

**Figure 14.** Bias and mean absolute error (MAE) of ICON-D2-ART compared to CERES SSF level 2 radiative fluxes for 1-5 March 2021. Shown are the outgoing longwave radiation at the top of atmosphere (TOA), the reflected shortwave at TOA and the solar irradiance at the surface (from top to bottom)





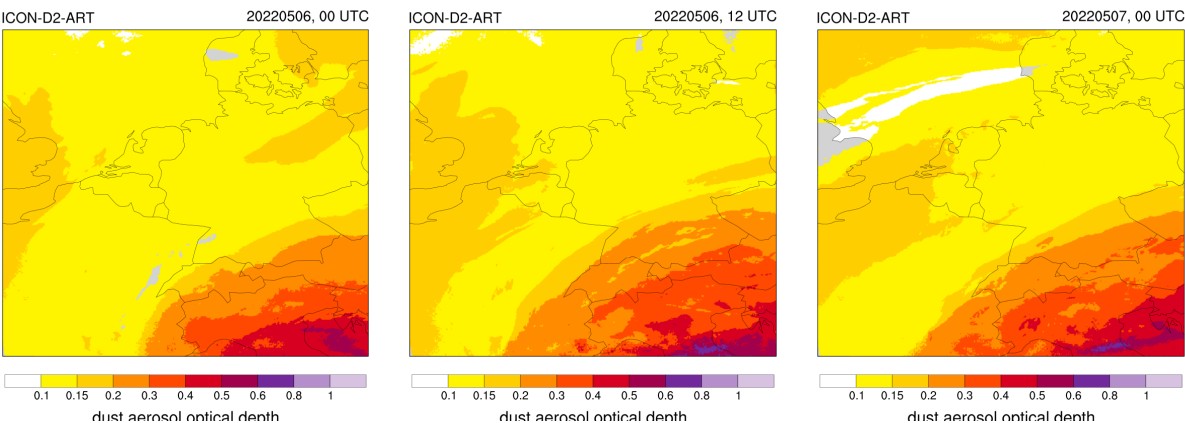

**Figure 15.** ICON-D2-ART mineral dust aerosol optical thickness (AOD) on 6 May 2022, 00 and 12 UTC, and 7 May 00 UTC.

the outgoing longwave radiation is reduced almost to zero for ACI-dusty, but significant biases remain for the shortwave fluxes due to issues with low clouds in ICON during this period. Validation of the case with radiosondes shows very similar results as in the case of 15-20 March 2022 and is discussed in the Supplement.

### 3.3   Dusty cirrus case of 4-8 May 2022

The Saharan dust case of 4-8 May 2022 differs from the two previous cases in that the dust transport is more to the east and

dust is advected from North Africa to Italy and the larger Alpine region. In the ICON-D2-ART domain, only the south-eastern part is affected by high dust AOD, which is rather persistent over several days (Figure 15). The dust event is associated with extended cirrus clouds, but in this case, all four simulations do have some cirrus cloud coverage over the Alps. In ACI-dusty the cirrus clouds are thicker and have a lower IR brightness temperature compared to ACI (Figure 16). Compared to the SEVIRI observations ACI-dusty significantly overestimates the cirrus cloud deck, and ACI underestimates the spatial extent but matches

the IR brightness temperature better than ACI-dusty. Hence, in this case, the dusty cirrus parameterization intensifies the cirrus deck in ICON-D2-ART but overestimates the impact of the dust. This might be due to the simple diagnostic nature of the scheme that is unable to represent the microphysical aging and dissipation of the cirrus cloud.

    For the individual Aqua overpass of 6 May, 12:55 UTC, shown in Figure 17 we find an overestimation of OLR in ACI over the Alpine region and an underestimation of OLR in ACI-dusty, and corresponding biases in the shortwave fluxes consistent

with an underestimation of the cirrus cloud in ACI and an overestimation in ACI-dusty.

    In the statistical evaluation of all CERES overpasses from 4-8 May 2022 shown in Figure 18 the ACI and ARI simulations are clearly superior to ACI-dusty. Hence, the dusty cirrus parameterization leads to an unrealistic intensification of the cirrus formation during this dust episode. The result is a significant negative bias in the OLR and increased biases in the shortwave fluxes compared to the other ICON simulations. This suggests that the dusty cirrus parameterization is too simple to describe all

dust events and the related cirrus clouds correctly. Given the simplicity of the parameterization, this is not surprising, though.





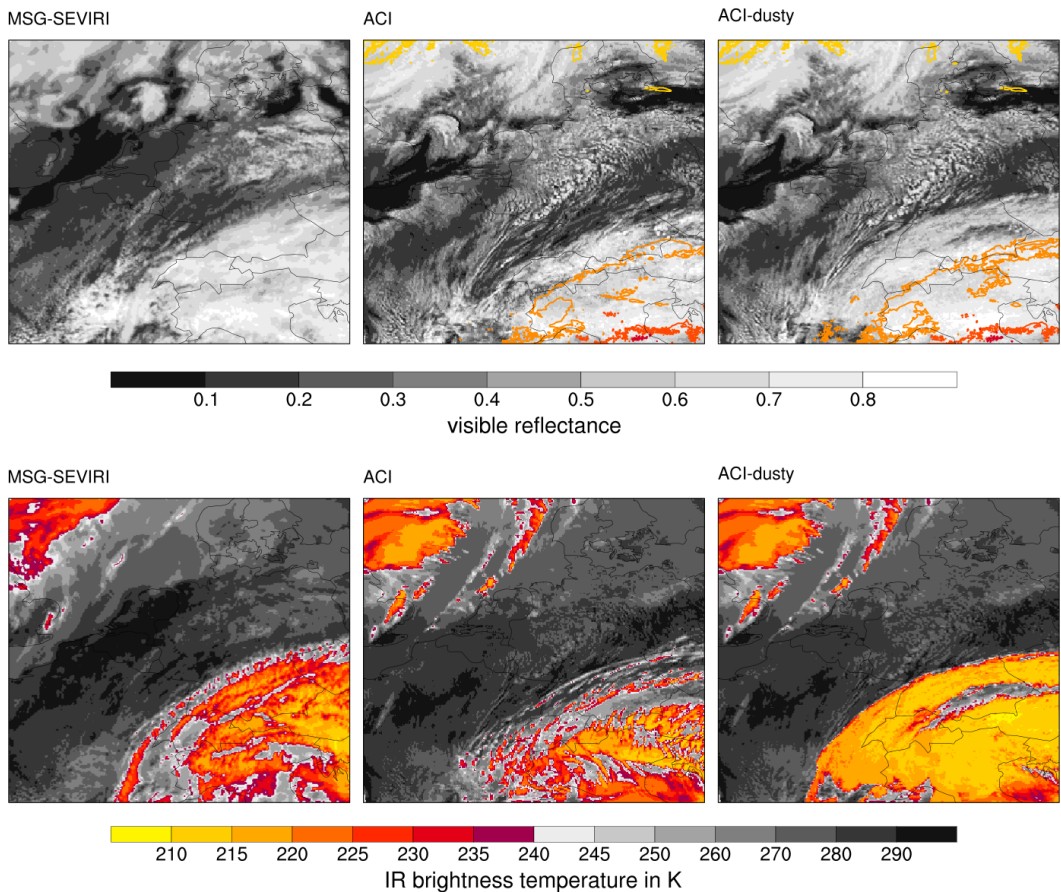

**Figure 16.** MSG SEVIRI visible reflectance (top) and infrared brightness temperature of the 10.8 $\mu$m channel (center) vs ICON-D2-ART on 6 May 2022, 12 UTC. The isolines in are the mineral dust AOD predicted by ICON-D2-ART (interval of 0.2 starting at 0.1)

The two main processes that are missing are, first, the transient behavior of the cirrus cloud depth during formation and decay of the cirrus deck and, second, the microphysical processes in the cirrus cloud, like aggregation and sedimentation. The lack of microphysical processes contributes to a prolonged lifetime and could cause the overestimation that we see in this case.

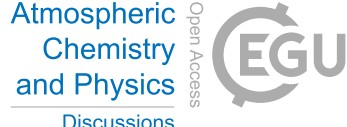

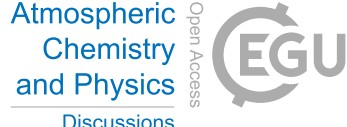

**Figure 17.** CERES SSF from Aqua vs ICON-D2-ART for 6 May 2022, 12:55 UTC.



a) ICON-D2-ART vs CERES outgoing longwave radiation at TOA

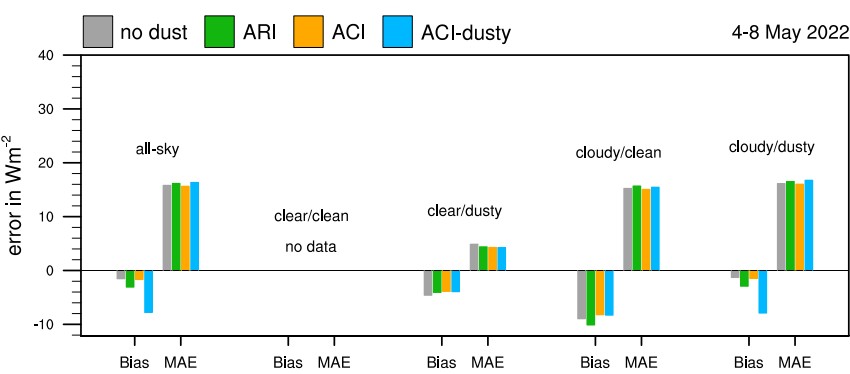

b) ICON-D2-ART vs CERES reflected shortwave radiation at TOA

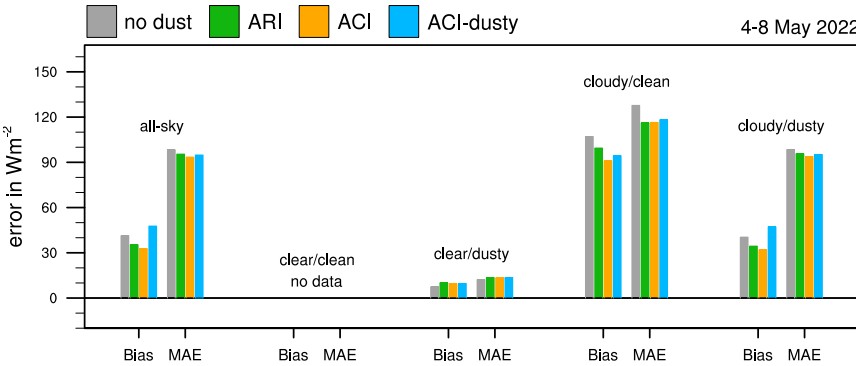

c) ICON-D2-ART vs CERES downward shortwave radiation at the surface

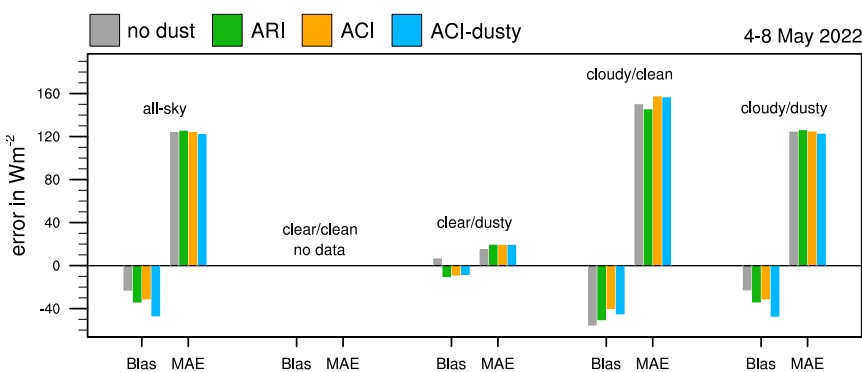

**Figure 18.** Bias and mean absolute error (MAE) of ICON-D2-ART compared to CERES SSF level 2 radiative fluxes for 4-8 May 2022. Shown are the outgoing longwave radiation at the top of atmosphere (TOA), the reflected shortwave at TOA and the solar irradiance at the surface (from top to bottom)





### 3.4 Statistical evaluation of six Saharan dust events

So far we have focused on case studies of Saharan dust events, which shows some observational evidence for dusty cirrus occurrence. For any practical application of the dusty cirrus parameterization, it is essential that it does not deteriorate the forecasts during dust events without dusty cirrus formation, i.e., the false alarm rate for dusty cirrus needs to be small. To check this, we have performed simulations for three more Saharan dust events without observational evidence for dusty cirrus (see Table 1). A detailed analysis of those individual events is given in the Supplement. Here we focus on the overall statistics
over all six Saharan dust events.

Figure 19 shows the Bias and mean absolute (MAE) errors for the radiative fluxes validated by CERES SSF data. For the solar irradiance at the surface, we see a clear improvement in clear/dusty conditions dominated by the direct radiation effect of mineral dust. Hence, the improvement is already seen in ARI without further gain from ACI or ACI-dusty. The impact of mineral dust in clear/clean conditions is small, as expected. For cloudy/dusty conditions we find an improvement by ARI
(because this includes partially cloudy pixels) and further improvements in MAE from ACI-dusty. The mean bias is smallest for ARI and ACI and becomes negative for ACI-dusty. But a close inspection reveals that the small mean bias for ARI and ACI is an error compensation between cases in early spring, which have a positive bias in those simulations, and late spring, which have a negative bias. This change in bias behavior is related to the occurrence of deep convection in May/June, which contributes to a negative bias due to the overestimation of convective anvils in ICON-D2. Hence, the small bias of ARI and
ACI should not be interpreted as a general forecast improvement. Similar arguments apply to OLR and reflected shortwave radiation at TOA. For both TOA fluxes we find the lowest MAE for ACI-dusty, but the mean bias becomes negative for OLR and increases for reflected shortwave radiation. Again, this is most likely related to an overestimation of deep convective anvils in ICON-D2, which leads to an error compensation in ARI and ACI. Hence, from this validation with CERES SSF level 2 data we find a small overall improvement from ACI-dusty compared to the other simulations but further work would be necessary
to, first, reduce the false alarms of the dusty cirrus parameterization and, second, improve the representation of convective anvils in ICON-D2.

The retrieval of solar irradiance at the surface from satellite data can lead to considerable uncertainties for cloudy pixels (Kratz et al., 2020). To further support our results we have performed an additional validation with the 27 pyranometers of DWD's radiation station network in Germany. Due to the rather short time periods of 5 days we refrain for a decomposition
in clear-sky and cloudy for the pyranometer data. As for the validation using CERES data, dusty conditions are defined based on ICON-D2-ART dust AOD exceeding 0.1. Figure 20 shows the results for the two major dusty cirrus episodes and for all 6 dust episodes investigated in this study. The biases and MAEs compare well with the results against CERES in Figs. 6, 14, and 19. This confirms not only our findings but also that CERES level 2 instantaneous fluxes can indeed be used for this kind of episode-based analysis. The results for the other dust episodes are provided in the Supplement.

More insight into aerosol-cloud-radiative effects can be gained from an analysis of the errors on individual CERES footprints. Figure 21 shows the (model-obs) errors as a function of the mineral dust AOD simulated by ICON-D2-ART. Hence, this analysis gives an indication whether the modeled radiative flux has the correct sensitivity to mineral dust AOD. The 'no dust'



a) ICON-D2-ART vs CERES outgoing longwave radiation at TOA

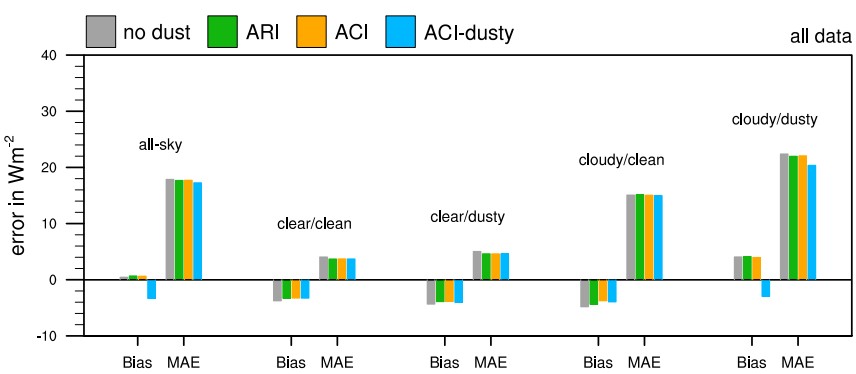

b) ICON-D2-ART vs CERES reflected shortwave radiation at TOA

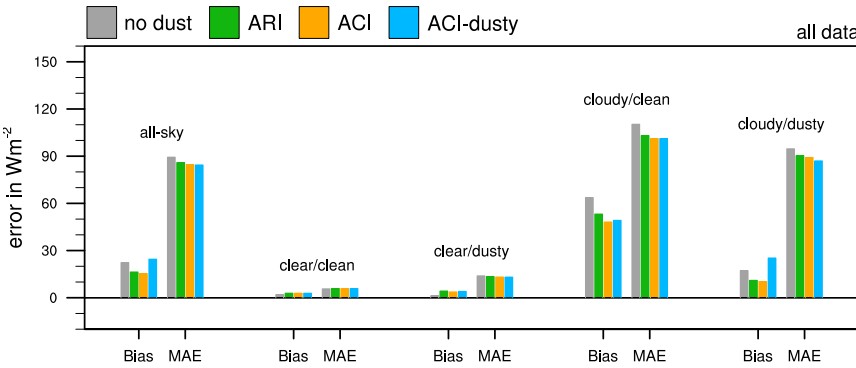

c) ICON-D2-ART vs CERES downward shortwave radiation at the surface

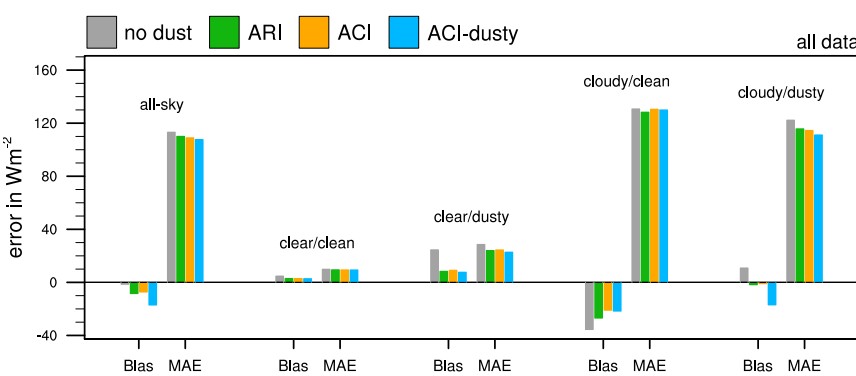

**Figure 19.** Bias and mean absolute error (MAE) of ICON-D2-ART compared to CERES SSF level 2 radiative fluxes for all six Saharan dust episodes. Shown are the outgoing longwave radiation at the top of atmosphere (TOA), the reflected shortwave at TOA and the solar irradiance at the surface (from top to bottom)





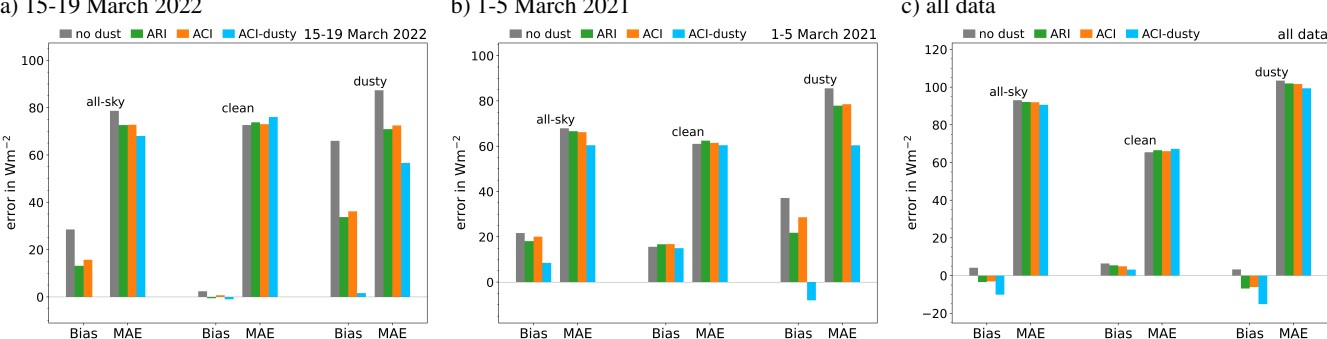

**Figure 20.** Bias and mean absolute error (MAE) of solar irradiance at the surface using DWD's pyranometer network for the dust episodes from 15-19 March 2022 (a), from 1-5 March 2021 (b), and for all data (c).

ICON-D2 simulation shows large positive errors for high dust AOD. This corresponds to the lack of direct and indirect aerosol effects during the Saharan dust episodes. A linear regression gives a slope of 336.8 Wm$^{-2}$ for the sensitivity $\Delta$SWD$_{sfc}$/AOD.

Taking into account the direct aerosol-radiative effect in ARI reduces these errors significantly and reduces the slope to 187.5 Wm$^{-2}$ but large errors remain especially for high AOD, which correspond to the occurrence of aerosol-cloud interaction. Considering grid-scale aerosol-cloud effects due to mineral dust acting as INPs has only a marginal impact in ICON-D2-ART and gives only a reduction of the slope to 175.6 Wm$^{-2}$. Only the newly developed sub-grid dusty cirrus parameterization is able to correct the large errors in solar irradiance for large AODs, which obviously correspond to the pixels below the extended

cirrus clouds decks. Interestingly, the slope $\Delta$SWD$_{sfc}$/AOD becomes almost zero, actually it is already slightly negative with a value of -12.1 Wm$^{-2}$. Similar behavior is found for OLR in Figure 22 but the impact of the direct aerosol is much weaker with a decrease in slope from 84.4 Wm$^{-2}$ for 'no dust' to 75.3 Wm$^{-2}$ for ARI. The improvement from ACI is again marginal with a slope of 71.3 Wm$^{-2}$, whereas the dust cirrus parameterization removes the dependency on mineral dust AOD resulting in a slope of -4.9 Wm$^{-2}$. A similar analysis of reflected shortwave radiation is given in the Supplement.

The fact that on one hand grid-scale ACI has only a negligible impact and, on the other hand, the dusty cirrus parameterization fully removes the aerosol sensitivity is remarkable and suggests that dusty cirrus formation is the dominant aerosol-cloud-radiative effect of mineral dust over Europe. This is somewhat unexpected because enhanced mineral dust concentrations during Saharan dust episodes should lead to more INPs being available for ice formation. It is often argued that high INP numbers increase the ice particle number concentration and subsequently increase ice water content and ice water path (e.g. Seifert et al.,

2012). Our simulations suggest that this sensitivity is very weak in ICON-D2-ART. A possible physical explanation is the so-called freezing-relaxation feedback, which is usually described for homogeneous ice nucleation (Kärcher and Seifert, 2016; Kärcher and Jensen, 2017), but which does in principle also apply to heterogeneous ice nucleation. Freezing-relaxation means that within an active updraft the ice supersaturation increases until enough ice particles are present to efficiently deplete the supersaturation. This establishes a nonlinear feedback or buffering mechanism, which can dramatically reduce the sensitivity

of the ice particle number concentration to INPs. Hence, a low number of INPs will lead to larger ice supersaturation, but not



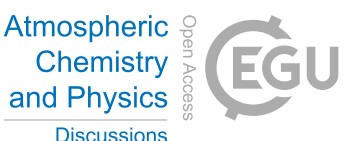

necessarily to a significant decrease in ice particle number concentration. Similarly, a high number of INPs, as during a Saharan dust event, will reduce the local supersaturations within the updrafts, but has little effect on the actual cloud properties like ice particle number, ice water content, or ice effective radius. This behavior is consistent with the recent findings of Chen et al. (2022), who showed that aerosol-cloud effects due to volcanic aerosol perturbations in low-level liquid clouds are dominated

by changes in cloud cover rather then by cloud brightening. Similarly, in our case the formation of the dusty cirrus and the corresponding increase in cloud cover dominates over changes in cloud optical properties due to mineral dust.

## 4 Summary and Conclusions

Outbreaks of Saharan dust reaching Europe are occasionally accompanied by the formation of a large-scale cirrus cloud, a phenomenon known as dusty cirrus. Since today's numerical weather prediction models neither predict mineral dust distributions

nor consider the interaction of dust with cloud microphysics, they cannot simulate this phenomenon. Among other things, this leads to significant forecast errors regarding photovoltaic power generation. Avoiding such forecast errors will become more and more important.

We have developed a simple subgrid-scale cloud parameterization scheme that allows simulating and therefore predicting dusty cirrus clouds. As the physical mechanism that leads to the formation of the dusty cirrus, we postulate a microphysical

mixing instability of clean moist and drier dusty air in the upper troposphere combined with synoptic-scale lifting of the air layers, e.g. through a WCB airstream underneath. Once a cirrus cloud has formed at the interface between moist and dusty air, the longwave cooling at cloud top generates turbulence and mixing which thickens the cloud layer.

The new dusty cirrus parameterization was included in ICON-ART and case and sensitivity studies were performed. ICON-ART allows calculating the interaction of mineral dust with radiation. When the two-moment cloud microphysics scheme is

used, also the interaction of mineral dust with grid-scale microphysical processes is calculated.

Although in one of the simulated cases, the dusty cirrus parameterization produced too thick, too extended cirrus layers and false alarms, we show that the new parameterization leads to a considerable improvement of the forecast of longwave and shortwave radiative fluxes at the top of the atmosphere and at the surface. Moreover, it leads to an improvement of the simulated cloud cover. In contrast, the two-moment microphysics scheme alone is not able to simulate the formation of the dusty cirrus,

although it is coupled to the prognostic mineral dust. In total six cases were simulated. In three of them, dusty cirrus occurred, and in three of them, mineral dust was present without dusty cirrus formation. Our new parameterization does not diminish the forecast quality in cases when dust outbreaks happened but no dusty cirrus occurred.

It remains to be proven that the concept of a microphysical mixing instability accompanied by an air mass configuration of moist air over dusty air applies not only to Europe but also to dusty cirrus formation in other parts of the world.



a) no dust

b) ARI

c) ACI

d) ACI-dusty

**Figure 21.** Scatter plots of errors in solar irradiance at the surface as function of mineral dust AOD predicted by ICON-D2-ART for all six Saharan dust episodes shown in Table 1. Shown is (model-obs) for individual CERES footprints with a diameter of 25 km. Colors indicate the absolute value of the observed solar irradiance.





**Figure 22.** As Figure 21 but for outgoing longwave radiation at top of atmosphere.



*Code availability.* The ICON code is available under two different licenses: A personal non-commercial scientific license, and an institutional license that requires a cooperation agreement with DWD. More details on the licenses and an instruction how to obtain the ICON code can be found at https://code.mpimet.mpg.de/projects/iconpublic.

*Author contributions.* A.S. performed the ICON-D2-ART simulations and carried out the evaluation using satellite and radiosonde data. A.S. developed and implemented the dusty cirrus parameterization. V.B. and J.F. maintain the ICON-ART model at DWD and performed the global ICON-ART simulations that serve as initial and boundary conditions for the current study. V.B. developed the ICON-D2-ART hindcast setup used in this study. B.V., G.A.H., H.V. and A.R. have developed the dust processes and the dust-radiation interactions in ICON-ART including the optical properties for ecRad as used in this study. F.F. performed the validation with DWD's pyranometer network, and A.W. provided the validation with DWD's ceilometer network. A.S. wrote the initial version of the manuscript. C.M.G and J.F.Q performed the trajectory analysis. All authors contributed to the interpretation of the results and the writing of the manuscript.

*Competing interests.* The authors declare that they have no conflict of interest.

*Acknowledgements.* A.S. is grateful to Bjorn Stevens, Jens Reichardt, Juan Pedro Mellado, Alberto de Lozar and Günther Zängl for helpful discussion. We thank Ivan Smiljanić of EUMETSAT for pointing us to the dusty cirrus case of 6 May 2022. We thank Annika Schomburg, Leonhard Scheck, Thomas Deppisch, Christina Stumpf, and Alberto de Lozar for their work on and help with the RTTOV forward operator. Annika Schomburg processed and provided the SEVIRI data for this study. Meteosat (MSG) SEVIRI data were obtained from EUMETSAT. CERES SSF data were obtained from NASA Langley Research Center (LaRC) Atmospheric Sciences Data Center (ASDC) at https://asdc. larc.nasa.gov/data/CERES/SSF (link requires Earthdata login). All simulations were performed on the NEC Aurora of DWD. Most plots were created with The NCAR Command Language (Version 6.6.2), UCAR/NCAR/CISL/TDD, http://dx.doi.org/10.5065/D6WD3XH5. This work contributes to and is partly funded by the project PermaStrom within the 7th Energieforschungsprogramm (funding code 03EI4010A) of the German Federal Ministry of Economic Affairs and Climate Action (Bundesministerium für Wirtschaft und Klimaschutz, BMWK). The contributions of CMG und JFQ are funded by the Helmholtz Association as part of the Young Investigator Group "Sub-seasonal Predictability: Understanding the Role of Diabatic Outflow" (SPREADOUT, grant VH-NG-1243).





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
