# Peer review of "Aerosol-cloud-radiation interaction during Saharan dust episodes: The dusty cirrus puzzle"

_Atmospheric Chemistry and Physics, 2022_

## Referee Comment (RC1)

This study developed and implemented a sub-grid parameterization for dusty cirrus over Europe in ICON-D2-ART model. The authors tested the parameterization during six Saharan dust episodes over Europe. It turns out that this sub-grid parameterization performs remarkably well in two of the selected cases. Overall, this is an innovative and interesting study. It fits well within the scope of ACP. However, I have concerns regarding the representativeness of the selected six cases. Also, there are too many figures in the main text. I have some specific comments as well.

General comments

1. Why are these six cases selected? How do the first three cases in Table 1 represent the dusty cirrus phenomena in Europe? More importantly, how do the last three cases represent the dust events with no occurrence of dusty cirrus? Specifically, how many dust episodes occur in one year in Europe? How many of these dust events have dusty cirrus and how many of them does not?

2. Analysis in Section 3.4 is performed using the six selected dust events, half of which are dusty cirrus cases. Following the above questions, I have concerns on the representativeness of the six cases for the yearly dust events in Europe. Dusty cirrus occurs in half of the six cases. However, it seems dusty cirrus is a rare phenomenon in Europe and thus, presumably, there are much more dust events that do not accompany with dusty cirrus. Therefore, it is not fair to evaluate the overall performance based on the six selected cases. Ideally, the authors should analyze all the dust events in one year. Or at least, the authors should analyze the cases with dusty cirrus and without dusty cirrus separately. More no dusty cirrus cases will be needed to increase the representativeness of such cases.

3. There are too many figures (22 figures) in the main text. Please consider move some to the supplement. For example, you may shorten Section 3.2 and move the related figures to the supplement because this case has similar results to the first one.

Specific comments

1. Figures 1, 3-5, 11-13, and 15-17. To help readers better identify the regions of these maps, please add latitude and longitude tick markers.

2. Section 2.1 and 2.2. It is not clear enough until I read the first paragraph of Section 3 that the ICON-D2-ART model includes dust-cloud interaction on grid scale, and the dusty cirrus parameterization is a sub-grid parameterization. Please clarify it. For example, reword the title of section 2.2 to "A sub-grid parameterization of dusty cirrus".

3. Line 68-69. Please explain which mode has the smallest size and which one has the largest size here. If possible, please give a rough estimate of the size range for each mode.

4. Eq (3) and Line 144-147. DeMott et al. (2010 and 2015) parameterizes ice nucleating ability using aerosol/dust number concentrations. Some other studies describe INP concentration based on dust surface area (e.g., Ulrich et al. 2017). By using dust mass concentrations in this study, the increased ice nucleating ability of dustC mode is already naturally considered, because coarser dust particles contribute more to total mass. I have no problem with further doubling the weight for dustC, but please explain it more clearly.

5. Table 2. How is dust treated in the no dust simulation? What does climatological dust mean? If it does include dust representation, the name "no dust" may be misleading.

6. Line 279 and Line 281: "ice condensation nuclei". Should be "INPs".

7. Line 322. What is microphysical aging and how does it contribute to the bias?

8. Figure 6. Please explain the differences between bias and MAE. It can be included either in the figure caption or the main text.

9. Line 381-382: "dusty cirrus formation is the dominant aerosol-cloud-radiative effect of mineral dust over Europe", and same statement appearing in the abstract. Still related to the representativeness issue mentioned in my general comments, it is not safe to make such statement, because it is very likely that you do not include all the dust-cloud interaction cases.

10. Line 380-385: "The fact that... very weak in ICON-D2-ART". This part is not clear to me. Figures 21 and 22 show model biases as a function of dust optical depth. I agree that the sub-grid parameterization reduces model biases over all the dust loading. But it seems the absolute radiation fluxes (colors of the scatters) do decrease with dust optical depth. Then, why is it concluded that the new parameterization removes aerosol sensitivity?

11. Figures S3, S4, and S7. Are these two figures identical to Figures 6, 14 and 18, respectively? If so, please remove these two figures in the supplement. If these is any other duplicate figure in the supplement, please remove them as well.

12. Figure S61: caption. Please confirm whether it is May 15 or May 5.

---

## Author Comment (AC1)

**Response to Reviewer 1**

We thank the reviewer for this thorough review, which helps us to improve the manuscript.

**Reply to general comments:**

**1. Why are these six cases selected? How do the first three cases in Table 1 represent the dusty cirrus phenomena in Europe? More importantly, how do the last three cases represent the dust events with no occurrence of dusty cirrus? Specifically, how many dust episodes occur in one year in Europe? How many of these dust events have dusty cirrus and how many of them does not?**

We have tried to select representative mineral dust episodes from the last two years. First of all, those three that showed clear indications of a dusty cirrus formation, i.e. optically-thick cirrus associated with mineral dust that was missed by standard global operational NWP systems like IFS or ICON. Then we tried to find three mineral dust episodes of similar intensity in terms of dust aerosol optical depth, which serve as control cases. The control cases are needed to make sure that our new parameterization can distinguish between mineral dust events with and without dusty cirrus formation. We think that this approach is actually quite fair and an attempt to test the model for representative cases.
It is not possible with the current DWD system to simulate older episodes, before 2021, with a consistent model setup. Including older cases would either require to use an older model version of ICON or accept significant inconsistencies in the initial condition (which deteriorates forecast skill and makes results questionable). Hence, we decided to stick to cases from the years 2021 and 2022. However, there are many interesting dust episodes in those two years, which allows us to have six cases in this paper.
An in-depth study of the climatology of the dusty cirrus phenomenon has, to our knowledge, not been performed for Europe or any other place. The colleagues at EUMETSAT have collected many cases based on satellite images, but those were never compiled into a scientific publication. Hence, we cannot answer the question how many cases of dusty cirrus occur on average over Europe.
The number of mineral dust episodes is somewhat easier to quantify. Table 1 of this reply gives an overview of the mineral dust episodes in 2021 and 2022 (up to and including July, when started writing the manuscript). The list of episodes is based on the operational German ceilometer network. The 2nd and 3rd column are the intensity and the maximum height of the dust layer based on the ceilometer backscatter data. Colums 5-7 are from five AERONET stations in Germany (Hohenpeissenberg, Lindenberg, Leipzig, Munich and Mainz). Given is the maximum coarse mode AOD at any of the 5 stations and the corresponding values of total AOD and Angstrom exponent (440-870 nm). The ICON-ART maximum dust AOD is the maximum of the simulated AOD of the global ICON-ART dust forecasting system taken at 9 Ceilometer stations in Germany (Hohenpeissenberg, Lindenberg, Leipzig, Weihenstephan,

| Period based on Ceilometer data | Intensity based on Ceilometer data | Dust Max Height [km] | ICON-ART max DAOD | AERONET total AOD | AERONET max coarse AOD | AERONET min Angstrom Exp. |
|---|---|---|---|---|---|---|
| **2021** | | | | | | |
| 05.02.2021-06.02.2021 | medium/ extreme | 5 | 0.55 | missing | missing | missing |
| **21.02.2021-26.02.2021** | **medium/ strong** | **8** | **0.97** | **0.64** | **0.57** | **0.02** |
| **02.03.2021-04.03.2021** | **strong** | **8** | **0.62** | **0.56** | **0.38** | **0.43** |
| 08.03.2021-09.03.2021 | weak | 8 | 0.11 | missing | missing | missing |
| 28.03.2021-30.03.2021 | weak | 6 | 0.21 | 0.16 | 0.13 | 0.25 |
| 31.03.2021-02.04.2021 | medium | 6 | 0.30 | 0.65 | 0.44 | 0.39 |
| 04.04.2021-05.04.2021 | weak | 8 | 0.27 | 0.30 | 0.15 | 0.88 |
| 17.04.2021-21.04.2021 | medium | 7 | 0.24 | 0.29 | 0.16 | 1.00 |
| **17.06.2021-21.06.2021** | **medium/ strong** | **5** | **0.44** | **0.50** | **0.38** | **0.37** |
| 11.07.2021-14.07.2021 | medium | 5 | 0.41 | 0.35 | 0.22 | 0.64 |
| 16.07.2021-20.07.2021 | weak | 9 | 0.17 | 0.26 | 0.14 | 0.79 |
| 23.07.2021-25.07.2021 | weak | 5 | 0.22 | 0.57 | 0.29 | 1.03 |
| 25.09.2021-26.09.2021 | medium | 5 | 0.22 | missing | missing | missing |
| **2022** | | | | | | |
| **15.03.2022-19.03.2022** | **extreme** | **10** | **0.83** | **missing** | **missing** | **missing** |
| 20.03.2022-25.03.2022 | weak/ medium | 10 | 0.26 | missing | missing | missing |
| 28.03.2022-30.03.2022 | medium | 10 | 0.36 | missing | missing | missing |
| 12.04.2022-14.04.2022 | strong | 6 | 0.44 | 0.57 | 0.44 | 0.16 |
| 18.04.2022-21.04.2022 | weak | 8 | 0.33 | 0.15 | 0.11 | 1.23 |
| **27.04.2022-03.05.2022** | **medium** | **4** | **0.33** | **0.47** | **0.24** | **0.60** |
| **04.05.2022-09.05.2022** | **medium** | **7** | **0.36** | **0.41** | **0.15** | **1.45** |
| 15.05.2022-16.05.2022 | weak | 4 | 0.34 | 0.21 | 0.11 | 0.78 |
| 18.05.2022-20.05.2022 | weak | 8 | 0.43 | 0.17 | 0.13 | 0.34 |
| 22.05.2022-23.05.2022 | weak | 5 | 0.50 | missing | missing | missing |
| 03.06.2022-05.06.2022 | weak | 6 | 0.26 | 0.24 | 0.20 | 0.29 |
| 15.06.2022-16.06.2022 | weak | 4 | 0.15 | missing | missing | missing |
| 17.06.2022-21.06.2022 | medium | 7 | 0.43 | 0.70 | 0.50 | 0.06 |
| 22.06.2022-23.06.2022 | weak | 4 | 0.24 | missing | missing | missing |
| 26.06.2022-30.06.2022 | weak | 7 | 0.30 | 0.31 | 0.24 | 0.44 |
| 03.07.2022-04.07.2022 | weak | 5 | 0.11 | missing | missing | missing |
| 17.07.2022-23.07.2022 | medium | 7 | 0.17 | 0.33 | 0.12 | 1.29 |

Figure 1: List of mineral dust episodes based on selected observations from DWD's operational ceilometer network over Germany and AERONET stations. The 6 cases of the manuscript are marked with yellowish color and bold font. The reddish color indicates events with a minimum intensity level of medium, based on ceilometer data (see text for details).

Mannheim, Würzburg, Stuttgart, Schleswig, Karlsruhe).

This list of dust episodes shows that the six cases presented in the paper are good candidates for interesting aerosol-cloud events and the occurrence of a 'dusty cirrus'. Indicators are high AOD and high maximum dust height. We could have exchanged the 27 April 2022 to 1 May 2022 with the event 28-30 March 2022, which has a deeper dust layer, or with 12-14 April 2022. We decided for the April-May episode, because we had some indications for aerosol-cloud-radiation effects for this case that we did not see for the other events. This is why we opted for that case. Most other mineral dust cases were either too weak or too shallow to be relevant for this study.

**2. Analysis in Section 3.4 is performed using the six selected dust events, half of which are dusty cirrus cases. Following the above questions, I have concerns on the representativeness of the six cases for the yearly dust events in Europe. Dusty cirrus occurs in half of the six cases. However, it seems dusty cirrus is a rare phenomenon in Europe and thus, presumably, there are much more dust events that do not accompany with dusty cirrus. Therefore, it is not fair to evaluate the overall performance based on the six selected cases. Ideally, the authors should analyze all the dust events in one year. Or at least, the authors should analyze the cases with dusty cirrus and without dusty cirrus separately. More no dusty cirrus cases will be needed to increase the representativeness of such cases.**

Including additional non-dusty cirrus cases, will not necessarily improve the quality of the current study. The additional dust episodes from 2021 and 2022 would have low to moderate dust AOD with dust located mostly at mid-levels. Hence, those will be well below the thresholds of the dusty cirrus parameterization suggested in the manuscript. We do not claim that our study is statistically representative for the full climatology of European mineral dust events. This would require simulations over several years, probably even several decades, to capture the variability of rare events. In our paper, we present only the statistics of the cases we have simulated. Yes, ideally we should have more cases, but we think that interesting hypotheses can be formulated based on the cases that are presented in the paper. This will hopefully foster additional research along the lines of the reviewer's very legitimate questions.

**3. There are too many figures (22 figures) in the main text. Please consider move some to the supplement. For example, you may shorten Section 3.2 and move the related figures to the supplement because this case has similar results to the first one.**

Yes, there are indeed many figures. We think they are interesting and important to support our hypothesis, but we will consider removing some that are not essential and somewhat redundant.

**Reply to specific comments:**

**1. Figures 1, 3-5, 11-13, and 15-17. To help readers better identify the regions of these maps, please add latitude and longitude tick markers.**

Unfortunately, ICON-D2 uses a rotated lat-lon domain. Hence, Figs. 3 and 4 are not in standard geographical coordinates. Adding regular latitude/longitude tickmarks is therefore not possible. It would be possible in Fig. 5 and some other plots and we will consider this for the revised version.

**2. Section 2.1 and 2.2. It is not clear enough until I read the first paragraph of Section 3 that the ICON-D2-ART model includes dust-cloud interaction on grid scale, and the dusty cirrus parameterization is a sub-grid parameterization. Please clarify it. For example, reword the title of section 2.2 to "A sub-grid parameterization of dusty cirrus".**

We have reworded the section title accordingly. That ICON-D2-ART includes grid-scale aerosol-cloud effects should become clear in section 2.1, though, were the aerosol-cloud coupling with the INAS-based ice nucleation schemes is described.

**3. Line 68-69. Please explain which mode has the smallest size and which one has the largest size here. If possible, please give a rough estimate of the size range for each mode.**

Typical median diameters with respect to the assumed particle number size distribution for the three ICON-ART dust modes are 0.6 $\mu$m for dustA, 2 $\mu$m for dustB, and 4 $\mu$m for dustC. These differ for dustB and dustC from the median diameters assumed in the emission scheme (dustA: 0.6 $\mu$m, dustB: 3.5 $\mu$m, dustC: 8.7 $\mu$m). The average median diameters in the atmosphere are smaller than the emission sizes, because large particles are removed rather quickly by sedimentation.

**4. Eq (3) and Line 144-147. DeMott et al. (2010 and 2015) parameterizes ice nucleating ability using aerosol/dust number concentrations. Some other studies describe INP concentration based on dust surface area (e.g., Ullrich et al. 2017). By using dust mass concentrations in this study, the increased ice nucleating ability of dustC mode is already naturally considered, because coarser dust particles contribute more to total mass. I have no problem with further doubling the weight for dustC, but please explain it more clearly.**

We make use of the INAS-based ice nucleation scheme of Ullrich et al. (2017). Hence, for (grid-scale) ice nucleation total dust surface area including the small dust particles is used. Our numerical experiments suggest that the formation of the dusty cirrus is dominated by large

dust particles and the smallest dust particles are rather unimportant. Hence, this would be somewhat more consistent with DeMott et al. (2010) and DeMott et al. (2015) rather than the original formulation of Ullrich et al. (2017). This argument for the preference for dustB and dustC (based on DeMott el al.) as predictors for dusty cirrus is already given in the manuscript. The fact that we give more weight to dustC than to dustB can be explained by the fact that dustC contains not so many small particles. One argument could be that the large particles of dustC have more cracks and pores and a higher INAS density than the other modes as discussed, e.g., by Holden et al. (2021). However, the details of ice nucleation in the atmosphere are still not sufficiently understood. For now, we see Eqs. (1)-(4) of the manuscript as a purely empirical result based on our simulations.

**5. Table 2. How is dust treated in the no dust simulation? What does climatological dust mean? If it does include dust representation, the name "no dust" may be misleading.**

As most operational global NWP systems, ICON uses a dust climatology in the control simulation, which is called 'no dust' for short. By dust climatology we mean that the dust concentration, or more precisely the dust optical thickness, is prescribed by monthly mean values. In ICON we still use the Tegen et al. (1997) aerosol climatology. Those monthly means are rather low for mineral dust and at least one order in magnitude smaller than the actual dust AOD during a significant dust episode. Hence, although the 'no dust' actually stands for 'no prognostic dust', given that the monthly mean dust AOD is so small, it is not completely wrong to think of these simulation as having no dust. We will consider replacing the labels by 'no prog. dust', though.

**6. Line 279 and Line 281: "ice condensation nuclei". Should be "INPs".**

Thanks, we have rephrased this.

**7. Line 322. What is microphysical aging and how does it contribute to the bias?**

By microphysical aging we mean any change in the size distribution or properties of cirrus ice particles that happens over time due to microphysical processes. For example, depositional growth, aggregation, and a concomitant transformation in particle habits. This will, over time, lead to an increase in the average sedimentation velocity of ice particles and a dissipation of the cirrus layer due to loss of particles. The diagnostic dusty cirrus parameterization does not take into account processes like aggregation and sedimentation. This may contribute to the bias with high IWC and a too optically thick cirrus layer, especially seen for the 6 May 2022 case (e.g. Figure 16).

**8. Figure 6. Please explain the differences between bias and MAE. It can be included either in the figure caption or the main text.**

The bias is also known as mean error (ME) and is different from the mean absolute error (MAE). This is rather basic statistics and we assume that readers of our paper would be familiar with this. We will consider to include a sentence somewhere that the bias is equivalent to the mean error.

**9. Line 381-382: "dusty cirrus formation is the dominant aerosol-cloud-radiative effect of mineral dust over Europe", and same statement appearing in the abstract. Still related to the representativeness issue mentioned in my general comments, it is not safe to make such statement, because it is very likely that you do not include all the dust-cloud interaction cases.**

The full sentence in the abstract reads: 'This suggests that the formation of dusty cirrus clouds is the dominant aerosol-cloud-radiation effect of mineral dust over Europe.' and, hence, we have weakened this statement by the cautious formulation 'This suggests... ' (in contrast, for example, to 'This proves...' or 'This shows...' which would be too strong here). In this moderate formulation it is, in our opinion, legitimate to make such a statement. The same applies to the sentence in the text, also here we say 'suggests that'. Of course, this requires that the sentence is quoted correctly.
The goal of our paper is mostly to provide evidence for the existence of the dusty cirrus phenomenon and to give some examples that a rather simple parameterization can be helpful to predict this dust-induced cirrus cloud. We also hope to encourage more research into the understanding of the dusty cirrus. Therefore we think that such statements are very useful, because they can be tested by follow-up studies.

**10. Line 380-385: "The fact that... very weak in ICON-D2-ART". This part is not clear to me. Figures 21 and 22 show model biases as a function of dust optical depth. I agree that the sub-grid parameterization reduces model biases over all the dust loading. But it seems the absolute radiation fluxes (colors of the scatters) do decrease with dust optical depth. Then, why is it concluded that the new parameterization removes aerosol sensitivity?**

The parameterization removes forecast errors in the aerosol sensitivity, it does of course not remove the aerosol sensitivity itself. In other words, the ACI-dusty simulation has an improved representation of aerosol sensitivity (of radiative fluxes) and by this the forecast error is reduced. As a result the forecast errors become independent from the dust AOD. We will consider to remove the coloring from these plots, because it seems to be rather confusing and distracting for some people. Of course, the absolute radiative fluxes at the surface have to decrease with dust AOD due to the direct and indirect aerosol effects of dust.

**11. Figures S3, S4, and S7. Are these two figures identical to Figures 6, 14 and 18, respectively? If so, please remove these two figures in the supplement. If these is**

**any other duplicate figure in the supplement, please remove them as well.**

Will be removed.

**12. Figure S61: caption. Please confirm whether it is May 15 or May 5.**

This is 5th of May 2022. Thanks for taking the time to look at the supplemental material.

**Response to Reviewer 2**

We thank the reviewer for this encouraging review, which helps us to improve the manuscript.

**Reply to general comments:**

**Figure 1 does not include the shortwave radiative effect. How does the shortwave heating rate at the thin ice cloud base/top altitude influence the destabilization effect?**

The cloud dynamics of the dusty cirrus and the destabilization at the moist-dusty interface is very similar to a stratocumulus cloud top. It is well known, that heating due to shortwave radiative effects is negligible in such a situation and the forcing is dominated by longwave cooling and evaporation/sublimation (e.g. de Lozar and Mellado, 2015; Mellado, 2017).

**The author assumed the same empirically determined thresholds for different dusty-cirrus events over Europe. How will these constant threshold values considered for different cases affect the testing of the sub-grid parameterization? Could you apply the same threshold condition for a similar dusty cirrus event in Asia?**

As we elaborate in section Conclusions, it is not at all clear that the dusty cirrus phenomenon as described and discussed in the manuscript does occur in other parts of the world. This is subject to future studies. Given the differences in continentality and air masses it is quite possible that the formation mechanisms for dust-induced cirrus in Asia are distinctly different from Europe. Therefore, it is curently not clear whether the parameters used in our study can be applied to other parts of the world.

**Besides the radiative part, what is the role of evaporative cooling in maintaining the dusty cirrus deck aloft?**

Following our hypothesis, evaporation cooling (or 'sublimative cooling') is essential for the dynamics of the dusty cirrus layers. Again, we would make use to the analogy to stratocumulus regime in which longwave radiative cooling and evaporative cooling are the main forcing of turbulence and circulation.

**Reply to specific comments:**

**Line 399f: Please rephrase this statement, "Since today's numerical weather prediction models neither predict mineral dust.....". There are regional models (e.g., the dust-coupled TAQM; Chen et al., 2004) which predict short-term 5-day forecasting of a dust event, including studying the dust effects on cloud microphysics using**

**TAQM-KOSA. A similar statement in the abstract can also be rephrased.**

We have rephrased this to 'today's global numerical weather prediction models'. The TAMQ dust model as described in Chen et al. (2004) would, in our opinon, be an environmental prediction model similar to CAMS or other chemistry-aerosol models. Those have usually a coarser grid compared to the regular NWP models and most often there is no feedback of the mineral dust on the dynamics through radiation or microphysics in these model.

**Line 406f: Instead of "Once a cirrus cloud has formed at the interface between moist and dusty air, the longwave cooling at cloud top generates turbulence and mixing which thickens the cloud layer.." I would suggest writing as "Once a cirrus cloud has formed at the interface between moist and dusty air, the dominant longwave cooling at cloud top generates turbulence and mixing which thickens the cloud layer."**

We have rephrased this using 'dominant longwave cooling' instead of just 'longwave cooling'.

**All the supplementary Figures can be referenced in the main text. The excess supplementary Figures can be taken out. If authors prefer to include all the supplementary Figures, they can include them in few .gif files or movies.**

We will considerably reduce the number of figures in the supplement.

**For clarity, the regional geographical images (Figures 1, 4-5, 11-12, and others) should include latitude and longitude labeling.**

We would like to do this, but due to the fact that most figures show a rotated lat/lon domain, this is unfortunately not possible. Therefore the continental boundaries and some national borders are shown to guide the eye. We admit that this is indeed a serious disadvantage of this kind of grid layout.

**Figure 6 caption should be elaborated. Please add a table or extend the caption to specify the "clear/clean" and "cloudy/dusty" terms.**

Those terms are introduced in section 3.1. We will add a short definition in the caption.

**Please add legends in Fig. 7c,d and Fig. 8a,b.**

We will consider to add a full legend to these plots.

**Quotation marks can be properly closed throughout.**

We will try to make sure that all quotations marks are properly closed.

**References**

A. de Lozar and J. P. Mellado. Mixing driven by radiative and evaporative cooling at the stratocumulus top. *J. Atmos. Sci.*, 72(12):4681–4700, 2015.

P. J. DeMott, A. J. Prenni, X. Liu, S. M. Kreidenweis, M. D. Petters, C. H. Twohy, M. Richardson, T. Eidhammer, and D. Rogers. Predicting global atmospheric ice nuclei distributions and their impacts on climate. *Proceedings of the National Academy of Sciences*, 107(25): 11217–11222, 2010. doi: 10.1073/pnas.0910818107.

P. J. DeMott, A. J. Prenni, G. R. McMeeking, R. C. Sullivan, M. D. Petters, Y. Tobo, M. Niemand, O. Möhler, J. R. Snider, Z. Wang, and S. M. Kreidenweis. Integrating laboratory and field data to quantify the immersion freezing ice nucleation activity of mineral dust particles. *Atmos. Chem. Phys.*, 15(1):393–409, 2015. doi: 10.5194/acp-15-393-2015.

M. A. Holden, J. M. Campbell, F. C. Meldrum, B. J. Murray, and H. K. Christenson. Active sites for ice nucleation differ depending on nucleation mode. *Proceedings of the National Academy of Sciences*, 118(18):e2022859118, 2021.

J. P. Mellado. Cloud-top entrainment in stratocumulus clouds. *Annual Review of Fluid Mechanics*, 49:145–169, 2017.

I. Tegen, P. Hollrig, M. Chin, I. Fung, D. Jacob, and J. Penner. Contribution of different aerosol species to the global aerosol extinction optical thickness: Estimates from model results. *J. Geophys. Res.*, 102(D20):23895–23915, 1997. doi: 10.1029/97JD01864.

R. Ullrich, C. Hoose, O. Möhler, M. Niemand, R. Wagner, K. Höhler, N. Hiranuma, H. Saathoff, and T. Leisner. A new ice nucleation active site parameterization for desert dust and soot. *J. Atmos. Sci.*, 74(3):699–717, 2017.

---

## Author Response (AR2)

**Response to Reviewer 1**

We thank the reviewer for this 2nd review, which helped us to further improve and clarify the manuscript.

**Reply to comments:**

**Regarding my general comment #1, the authors gave a convincing explanation about how the six cases (especially the three without dusty cirrus) are selected. I think it is worthwhile to mention in the manuscript that the three non-dusty cirrus cases were selected because they have similar dust intensity and height to the three dusty cirrus cases.**

We have included this a the beginning of section 3 and also in the Supplement.

**Regarding my general comment #2, I understand adding more non-dusty cirrus cases is not quite doable due to the limited number of available relevant cases, and I agree the current six cases are enough for this study. However, I still think the statistical evaluation in Section 3.4 does depend on how many non-dusty cirrus cases are included, since the new parameterization performs not as well for the non-dusty cirrus cases as for the dusty cirrus cases (Figures S5, S6, and S8). The improvements by ACI-dusty shown in Figures 19c and 20 may not be as evident with more non-dusty cirrus cases included. I would recommend the authors to add more discussion about this issue.**

We have included some note of caution about the robustness of the results in section 3.4.

**Regarding my specific comment #9, I understand that the statement is weakened by saying "This suggests . . . ". But my previous concern lies in the usage of "dominant" in the sentence. I agree with the authors that this study has proved the existence of dusty cirrus and the simple parameterization they proposed did an impressive job in simulating this phenomenon. However, as the authors mentioned in their responses to my general comment #2, this study is not statistically representation for the full climatology of European mineral dust events. Without examining the full climatology of dust events, I do not think it is safe to conclude (even with the word "suggest") that dusty cirrus is the dominant effect. I think Figure 20 only suggests that the dusty cirrus formation dominates over the other ACI interactions in these selected cases, when dusty cirrus presents and/or when the dust events are intensive and reach high altitude.**

We have changed the formulation to '.. for the six Saharan dust episodes investigated in the

current study ...' to avoid any generalization.

**Line 173: It seems Sections 3-8 in the previous supplement were deleted in the revised supplement. So, I am not sure which part in the supplement is this sentence referring to.**

Has been deleted.

**Line 341: "A detailed analysis of those individual events is given in the Supplement". Please give the figure numbers you are referring to.**

Has been deleted.

We have again checked all references to the supplement and corrected some mistakes, which occured during the revision of the manuscript.